**Mapping mining-affected water pollution in China: Status, patterns, risks, and**
**implications**
Ziyue Yin[1], Jian Song[2], Dianguang Liu[1], Jianfeng Wu[1,*], Yun Yang[2], Yuanyuan Sun[1], Jichun Wu[1]
[1] Key Laboratory of Surficial Geochemistry, Ministry of Education, Department of Hydrosciences,
School of Earth Sciences and Engineering, Nanjing University, Nanjing 210023, China
[2] School of Earth Sciences and Engineering, Hohai University, Nanjing 211100, China
[*] Corresponding authors. Tel: +86 25 89680853; fax: +86 25 83686016
*E-mail address*: jfwu@nju.edu.cn (J.F. Wu)

**Abstract:** Mining-affected water pollution poses a serious threat to human health and economic prosperity globally. The human toxicity and ecosystem impacts induced by mining activities have achieved considerable public, scientific, and regulatory attention. In this study, a comprehensive database of 8,433 water samples from 211 coal mines and 87 metal mines in China was established to reveal the national status and spatial heterogeneity of mining-affected water pollution, human health risks, and their potential multifaceted challenges. The results show that the concentrations of sulfate, Fe, Mn, Al, and several heavy metals in the mining-affected water of metal mines are generally higher than those of coal mines, especially in acid water (pH < 6.5). In terms of spatial distribution, the gridded data demonstrates that the southern regions in China, especially Guizhou, Guangdong, Fujian, Jiangxi, Hunan, and Guangxi provinces/autonomous regions, are the hotspots of mining-affected water pollution (*i.e.*, low pH as well as high sulfate, Fe, Mn, and heavy metals). The risk assessment reveals that the unacceptable carcinogenic risks caused by poor-quality surface water and groundwater are observed in 51.52% (for adults) and 29.29% (for children) of the mining areas. Moreover, severe non-carcinogenic risks are also identified in 68.07% and 80.67% of mining areas for adults and children, respectively. Overall, the acid and metal-rich water exhibits a widespread and detrimental impact in China, especially in the southern regions, posing significant risks to planetary health by degrading surface water and groundwater quality, destroying biodiversity, and threatening human well-being. This study provides a thorough set of scientific data on surface water and groundwater quality in mining areas to guide policymakers in designing differentiated management strategies for the sustainable development of coal and metal mines.

**Keywords:** Mining-affected water pollution; Spatial patterns; Risk assessment; Adverse effects; Differentiated management.

# 1 Introduction

Coal and metalliferous mineral resources are essential materials for global socio-economic development. The extraction and processing of minerals have caused detrimental impacts on aquatic ecosystems, soil ecosystems, living organisms, and human health worldwide (Blowes et al., 2014; Li et al., 2014; Havig et al., 2017; Ighalo et al., 2022). Mine drainage and leachate from active and abandoned mines are of global concern as they continue to release harmful substances into the underlying geological materials or adjacent water bodies for decades, inevitably leading to the degradation of both surface water and groundwater quality (Acharya and Kharel, 2020; Ighalo and Adeniyi, 2020). In particular, the environmental risks induced by acid mine drainage (AMD) have been ranked second only to global warming and ozone depletion (Moodley et al., 2018; Ai et al., 2023). Mining-affected water is generally characterized as metalliferous. Certain metals, such as Cu, Fe, Mn, and Zn, function as essential trace elements in human physiological processes. However, when their concentrations exceed specific thresholds in surface water and groundwater systems, these biologically relevant metals can pose significant toxicological risks to ecosystems and human health (Wei et al., 2022). Other non-essential heavy metals (HMs), including As, Cd, Cr, Hg, Ni, and Pb, have no nutritional or beneficial effects on humans. They can be toxic even at low concentrations and are therefore recognized as carcinogenic, mutagenic, and teratogenic. In addition, the persistence, toxicity, mobility, and non-biodegradability of HMs potentially form an enduring environmental footprint that jeopardizes ecosystems (He et al., 2013; Dippong et al., 2024). Consequently, there is a growing demand in mining areas for assessing the status of pollution and associated risks, as well as developing more effective management strategies and policies to mitigate these detrimental impacts (Cheng 2003; Hu et al., 2014).

Exploring the heterogeneity, risks, and threats of mining-affected water pollution is desirable
but remains challenging. More recently, an increasing number of studies have been focused on the
mining-affected water pollution from coal mines in major coal-producing countries (Sun et al.,
2013; 2020; 2025; Acharya and Kharel, 2020; Dong et al., 2022; Ai et al., 2023; Hou et al., 2024;
Kumar et al., 2024). For instance, Acharya and Kharel (2020) provided an in-depth overview of
the formation and effects of AMD from coal mining in the United States, reviewed prediction and
treatment methods, identified key research gaps, and explored the challenges and opportunities that
AMD posed for scientists and researchers. Ai et al. (2023) developed a conceptual model to
illustrate the formation and evolution of AMD in the coal mines from a life-cycle perspective.
Meanwhile, the critical governing factors and treatment technologies of AMD across abandoned
mines in major coal-producing countries were identified, including China, the United States, the
United Kingdom, Australia, and India. Coal and metal mines have different priority pollutants and
levels of pollution due to differences in geological conditions and mineral extraction methods (Yu
et al., 2024). Comparative studies of the status, heterogeneity, risks, and impacts of water pollution
in coal and metal mines achieved limited concerns so far. It is essential for the development of
remediation strategies and the implementation of risk-based, differentiated management practices
to achieve sustainability in mining areas associated with the mineral economy.
To our knowledge, previous studies have provided a solid basis for the soil pollution status of
HMs and their related health risk at the national or global scale (Li et al., 2014; Liu et al., 2020;
Hou et al., 2023; Shi et al., 2023). For example, Shi et al. (2023) identified the spatiotemporal
distribution of soil HM concentrations based on studies conducted between 1977 and 2020. In
addition, the ecological and human health risks were assessed concerning different land use types
at the national scale. Yu et al. (2024) provided a more comprehensive analysis of the pollution
characteristics, spatial distribution, major influencing factors, and probabilistic health risks of
potentially toxic elements in soil, using data from 110 coal mines and 168 metal mines across China.
However, systematic studies of water pollution status and risks have yet to be undertaken at a
national or even broader scale, as current research only focused on water pollution and risks in
specific mining areas (He et al., 1998; Xiao et al., 2003; Wang et al., 2019; Chen et al., 2020; Wang
et al., 2023). Therefore, it is necessary to implement deep mining of massive hydrochemical data
and establish a nationwide database that can identify the spatial pattern of mining-affected water
pollution and support risk assessment.
China, the second-largest economy worldwide, has various and extensive mineral resources
(Li et al., 2014). It has been demonstrated that there are 171 types of mineral resources in China,
with proven reserves accounting for 12% of the world's mineral resources (Hu et al., 2009).
Furthermore, China is one of the largest global producers and consumers of metals and metalloids,
such as Fe, Mn, Zn, Pb, Sb, and Sn (Gunson and Jian, 2001). China's coal reserves of 143,197
million tons (Mt) rank fourth globally, while its annual production of 2,971 Mt leads worldwide
(Blowes et al., 2014; Ai et al., 2023). The coal extraction inevitably generates substantial amounts
of mine water, resulting in a series of water environmental issues (Zhang et al., 2016b; Qu et al.,
2023). For example, Gu et al. (2021) demonstrated a 2:1 mine water to coal production ratio, with
approximately 2 tons of mine water produced per ton of extracted coal in China. In recent years,
China has put forward a series of monitoring, prevention, management, and remediation measures
to improve water quality and ensure water supply safety. However, the detrimental impacts
triggered by mining activities on the aquatic environment have not been well managed. Since 2010,

China has implemented a policy-driven initiative to phase out nearly 12,000 coal mines to address two critical challenges, *i.e.*, the declining economic viability and the escalating environmental externalities (Ma et al., 2020). These policies effectively restore water storage capacity in mining regions. However, sulfates and dissolved metals generated by complex geochemical processes during the weathering of sulfide minerals may migrate and transform within the recovering groundwater system, thereby increasing the ecological vulnerability of local hydrological networks.

Therefore, the objectives of the study are: (i) to establish a national-scale high-quality database containing basic water quality information for typical coal and metal mines; (ii) to reveal spatial heterogeneity of mining-affected water and evaluate health risks posed by potentially toxic elements from coal and metal mines drainage in China; and (iii) to highlight the negative impacts and discuss the management implications in the differentiated policy for different mine types (coal or metal) and multiple mining phases (active or abandoned). Exploring the spatial heterogeneity of mining-affected water in China is of great importance to achieve deep insights for designing the targeted and promising mitigation strategies at the different spatial scales, which is critical to implementing the optimal trade-offs between green mining and human health.

## 2 Data and methodology

### 2.1 Data mining and processing

The belief information of natural resources in China has been presented in Section S1 of the Supplement, which serves as the cornerstone for the database development, spatial pattern analysis, and risk assessment in the study. Specifically, Figs. S1 and S2 illustrate the spatial distribution and total sulfur content of coal-bearing areas in China, and Fig. S3 exhibits the spatial distribution of

the major non-ferrous mineral resources in China. In this study, the composite database integrates
mining-affected water (surface water and groundwater) quality parameters systematically extracted
from 293 peer-reviewed studies published over the past decades. The primary data were obtained
from mainstream online bibliographic databases, such as China National Knowledge Infrastructure,
China Wanfang Literature Database, Web of Science, Elsevier, Springer, Wiley, Taylor & Francis,
and the Multidisciplinary Digital Publishing Institute. The screening keywords were 'China', 'coal
mine', 'metal mine', 'acid mine drainage', 'mine water', 'surface water', 'groundwater',
'hydrochemistry', and 'heavy metals'. All retrieved literature was downloaded by 2024/4/25, and
the irrelevant studies were eliminated based on their abstracts, data, and full-text content.
*2.2 Quality assessment*
To ensure the reliability of the data, the collected literature was assessed for quality based on
the following criteria: (i) adhering to strict quality assurance/quality control procedures during
sampling, storage, and laboratory testing to ensure consistency, precision, and accuracy of results;
(ii) extracting the sampling year (if not stated, the received or published date of the manuscript was
adopted); (iii) extracting the latitude and longitude coordinates of the sampling site, mine or the
county-level city in which they are located; and (iv) extracting the concentration of the featured
component or statistics (minimum, mean and maximum value) based on the original data.
*2.3 Database establishment*
To assess the national extent of mining-affected water pollution, a comprehensive database of
8,433 data (6,175 coal mine data and 2,258 metal mine data) derived from 298 mines was
established, including 211 coal mines and 87 metal mines (*i.e.*, antimony mine, copper mine, gold
mine, hematite mine, iron mine, lead-zinc mine, molybdenum mine, polymetallic mine, pyrite mine,
rare earth mine, thallium-mercury mine, tin mine, tungsten mine, and uranium mine). The spatial
distribution of the sampling site used in the study and the data classification at the provincial level
are displayed in Fig. 1. The detailed information includes the sample ID, province, county/mine
name, latitude (N), longitude (E), mine type, mine status (active or abandoned), sampling year,
sampling month, sample type, basic physiochemical characteristics (pH, temperature (T), electrical
conductivity (EC), oxidation reduction potential (ORP), dissolved oxygen (DO), and total
dissolved solids (TDS)), major cation/anion ions ($Na^+$, $K^+$, $Ca^{2+}$, $Mg^{2+}$, $Cl^-$, $SO_4^{2-}$, $HCO_3^-$, $NO_3^-$ and
$F^-$), Fe, Mn, Al, HMs (Cr, Ni, Cu, Zn, As, Cd, Hg, and Pb) and data source. The typical mine lists
used in the study are shown in Table S1.

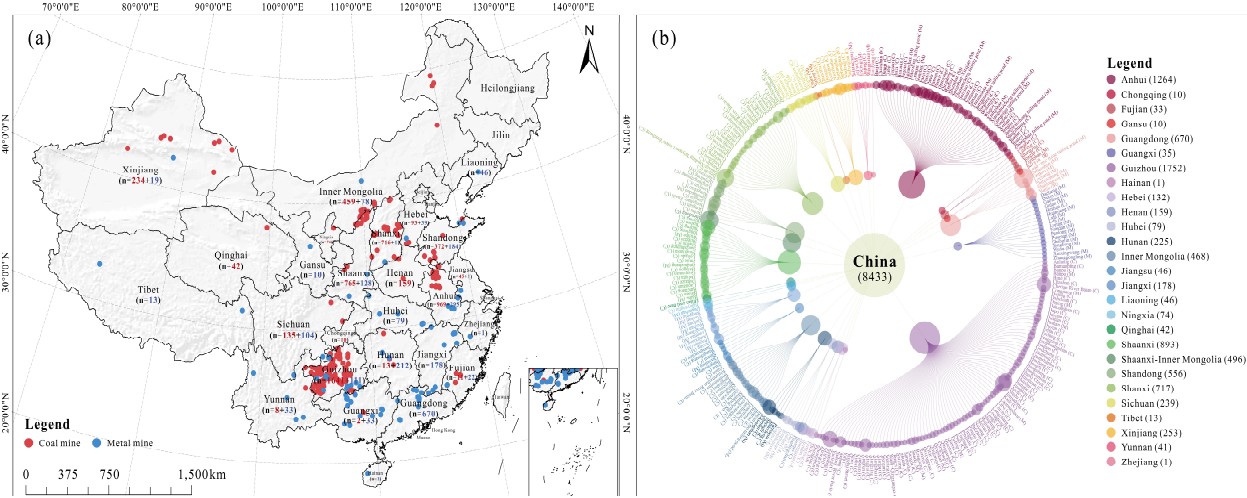


**Figure 1.** The information on data sources for this study, including (a) the spatial distribution of
the sampling site, and (b) the data classification at the provincial level. In Fig. 1a, the value in the
bracket represents the sample size (n), specially, the red and blue numbers are the sample sizes of
coal mines and metal mines for different provinces, respectively. In Fig. 1b, the size of the inner
circle represents the sample size at the provincial level, while the size of the outer circle represents
the sample size at the specific mine level (the letters 'C' and 'M' in the brackets represent coal mine
and metal mine, respectively). In the legend, the value in the bracket represents the sample size of
the different provinces.
*2.4  Risk assessment*
Human exposure to metals can occur through various pathways, including ingestion and
dermal contact with contaminated water. Therefore, these two pathways are considered to assess
the potential human risks, *i.e.*, non-carcinogenic risks (NCRs) and carcinogenic risks (CRs), for
adults and children. The model developed by the U.S. Environmental Protection Agency (US EPA)
is employed for risk assessment in this study (US EPA, 2004; 2011):
$$\mathrm{ADI_{ing}} = \frac{C_w \times IR \times EF \times ED}{BW \times AT} \tag{1}$$

$$\mathrm{ADI_{der}} = \frac{C_w \times K_p \times ET \times SA \times EF \times ED \times CF}{BW \times AT} \tag{2}$$

where $\mathrm{ADI_{ing}}$ and $\mathrm{ADI_{der}}$ are the average daily intake by ingestion and dermal adsorption (mg/kg·d),
respectively; $C_w$ is the metal concentration in the mining-affected water (mg/L); IR is the ingestion
rate (L/d); EF is the exposure frequency (d/yr); ED is the exposure duration (yr); $K_p$ is the
permeability coefficient of skin (cm/h); ET is the exposure time (h/d); SA is the exposed skin
surface area ($cm^2$); CF is the conversion factor ($L/cm^3$), which is set to 0.001 in the study; BW is
the body weight (kg); and AT is the averaging time (d).
The hazard quotient (HQ) and hazard index (HI) are used to determine the NCRs to human
health (Dippong et al., 2024). The HQ to residents (adults and children) from metal exposure via
ingestion ($\mathrm{HQ_{ing}}$) and dermal absorption ($\mathrm{HQ_{der}}$) are quantified:
$$\mathrm{HQ_{ing}} = \frac{\mathrm{ADI_{ing}}}{\mathrm{RfD_o}} \quad \text{and} \quad \mathrm{HQ_{der}} = \frac{\mathrm{ADI_{der}}}{\mathrm{RfD_{der}}} \tag{3}$$

$$HI = \sum HQ_i = HQ_{ing} + HQ_{der} \tag{4}$$

where HI is the hazard index, which is the sum of HQ. HI > 1 indicates potential adverse effects

on human health, whereas HI < 1 suggests no NCR is present; $RfD_o$ is the reference dose for oral

intake; and $RfD_{der}$ is the reference dose for dermal exposure, which can be calculated by:

$$RfD_{der} = RfD_o \times ABS_{GI} \tag{5}$$

where $ABS_{GI}$ is the gastrointestinal digestion coefficient (unitless).

The CRs to residents from ingestion and dermal absorption of mining-affected water are

determined using Eqs. (6) and (7):

$$CR_{ing} = ADI_{ing} \times SF \quad \text{and} \quad CR_{der} = ADI_{der} \times SF \tag{6}$$

$$TCR = \sum CR_i = CR_{ing} + CR_{der} \tag{7}$$

where TCR is the total CR, if TCR > $10^{-4}$, there is a significant risk to humans, and if $10^{-6} \le TCR$

$\le 10^{-4}$, the risk is generally acceptable; $CR_{ing}$ and $CR_{der}$ are the CRs induced by ingestion and dermal

contact with mining-affected water, respectively; SF is the slope factor. The detailed values of the

parameters in the above formula (Eqs. (1) - (7)) are represented in Tables S2 and S3.

## 3 Results and analysis

### 3.1 Overview of mining-affected water in China

All the data were collected from 26 provinces in China, while Beijing, Tianjin, Shanghai,

Heilongjiang, Jilin, Hong Kong, Macao, and Taiwan did not meet the data extraction principles. At

the provincial administrative level, five regions (*i.e.*, Guizhou, Anhui, Shaanxi, Shanxi, and

Guangdong) have a statistically significant representation in sample collection. Among these,

Guizhou and Anhui exhibit the most substantial data size, accounting for 20.78% and 14.99% of

the total national dataset, respectively (Fig. 1). The spatial distribution of the sample size for each
component at the 0.5° grid scale is depicted in Fig. S4. In general, most of the data show the basic
information of the water sample, such as pH value and the concentrations of major cation and anion
ions. In addition, many studies focused on the status of HM pollution in the mining areas, with a
multi-source research synthesis revealing substantial monitoring coverage of 2,241 (Fe), 2,265
(Mn), 1,401 (Al), 952 (Cr), 691 (Ni), 1,563 (Cu), 1,575 (Zn), 1,451 (As), 1,627 (Cd), 280 (Hg),
and 1,425 (Pb) water samples nationwide. Therefore, the Fe, Mn, Cd, Cu, and Zn are priority
investigative targets for mine water research. The statistics of acid and neutral/alkaline mining-
affected water in China are presented in Table 1. The pH and multi-component concentrations of
mining-affected water in China are shown in Fig. 2.

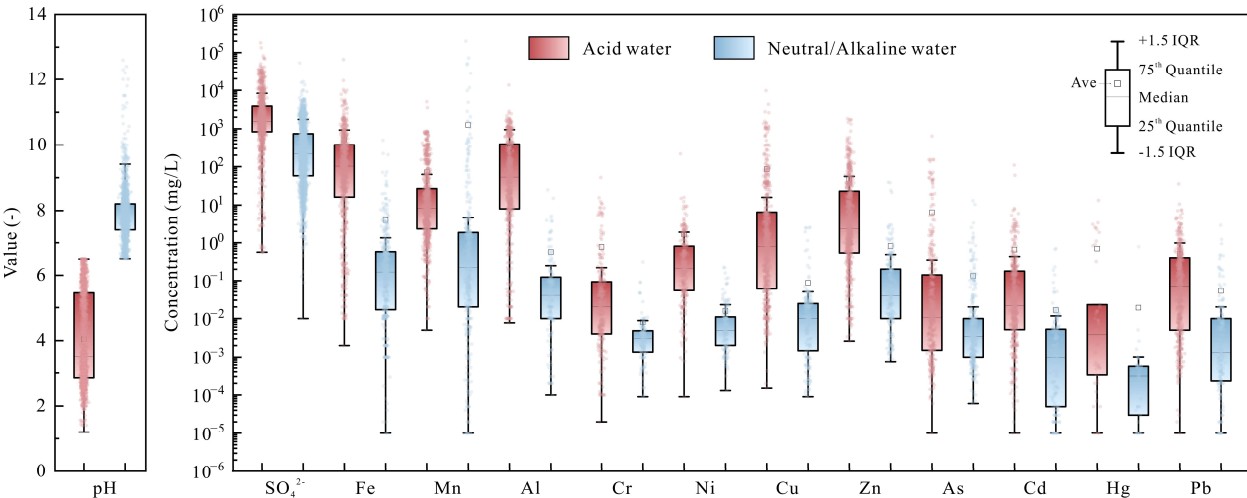


**Figure 2.** Statistical summary (minimum, median, average, and maximum) of the main species
aggregated from all samples measured in mining-affected water in China (Units are mg/L except
pH).

**Table 1.** Statistical summary (minimum, median, average, and maximum) of the main species aggregated from all samples measured in acid or non-acid mining-affected water in China (Units are mg/L except pH).

| Item | Acid mining-affected water | | | | Non-acid mining-affected water | | | |
|---|---|---|---|---|---|---|---|---|
| | Min | Median | Ave | Max | Min | Median | Ave | Max |
| pH | 1.20 | 3.52 | 4.04 | 6.50 | 6.51 | 7.80 | 7.85 | 12.60 |
| $Na^+$ | 0.00 | 15.60 | 55.39 | 1613 | 0.23 | 156.01 | 403.20 | 9371 |
| $K^+$ | 0.00 | 3.30 | 6.41 | 172.00 | 0.00 | 3.11 | 7.10 | 419.00 |
| $Ca^{2+}$ | 0.83 | 284.80 | 284.33 | 987.97 | 0.00 | 64.10 | 119.76 | 4841.70 |
| $Mg^{2+}$ | 0.01 | 75.33 | 388.36 | 10992 | 0.00 | 19.01 | 52.85 | 12752 |
| $Cl^-$ | 0.00 | 3.50 | 35.50 | 3097.40 | 0.00 | 51.68 | 314.21 | 26265 |
| $SO_4^{2-}$ | 0.56 | 1580.16 | 4648.54 | 181000 | 0.01 | 181.27 | 621.24 | 52915 |
| $HCO_3^-$ | 0.00 | 1.89 | 48.65 | 769.00 | 0.00 | 253.23 | 343.13 | 4976.61 |
| $NO_3^-$ | 0.00 | 0.80 | 16.46 | 735.60 | 0.00 | 3.78 | 16.02 | 1774.95 |
| $F^-$ | 0.00 | 0.69 | 4.17 | 238.34 | 0.00 | 0.81 | 1.91 | 100.00 |
| Fe | 0.0020 | 103.30 | 520.44 | 65250 | 0.00 | 0.1700 | 3.98 | 495.43 |
| Mn | 0.0050 | 8.11 | 71.18 | 5050 | 0.00 | 0.2164 | 1258.99 | 200000 |
| Al | 0.0077 | 53.90 | 304.92 | 13679 | 0.0001 | 0.0350 | 0.53 | 25.00 |
| Cr | 0.0000 | 0.0200 | 0.77 | 52.27 | 0.0001 | 0.0031 | 0.01 | 0.31 |
| Ni | 0.0001 | 0.2166 | 1.64 | 216.00 | 0.0001 | 0.0050 | 0.02 | 0.23 |
| Cu | 0.0002 | 0.8010 | 85.64 | 9777.77 | 0.0001 | 0.0100 | 0.07 | 2.56 |
| Zn | 0.0026 | 2.3685 | 46.63 | 1834 | 0.0007 | 0.0391 | 0.53 | 39.30 |
| As | 0.00 | 0.0108 | 6.50 | 641.70 | 0.0001 | 0.0034 | 0.14 | 13.00 |
| Cd | 0.00 | 0.0220 | 0.66 | 110.00 | 0.00 | 0.0004 | 0.01 | 0.67 |
| Hg | 0.00 | 0.0038 | 0.70 | 13.36 | 0.00 | 0.0003 | 0.02 | 0.78 |
| Pb | 0.00 | 0.0700 | 0.51 | 35.68 | 0.00 | 0.0012 | 0.03 | 1.22 |

It should be highlighted that the mean values may result in overestimation, as some extremely
high values are observed in the surveyed mines, such as the Baiyin copper mine, Bainiuchang
polymetallic mine, Zijinshan copper mine, and so on. Therefore, median values are selected to
represent the national characteristics of the mining-affected water pollution in this section. The pH
value of acid water (*i.e.*, pH < 6.5) ranges from 1.20 to 6.50, with a median (interquartile range,
IQR) of 3.52 (2.85, 5.45) (CV = 34.39%). In comparison, neutral/alkaline water has a pH value
between 6.51 and 12.60, with a median of 7.80 (IQR: 7.40, 8.20) (CV = 8.99%). Generally, the
$SO_4^{2-}$ concentration of acid water is higher than that of neutral/alkaline water (Figs. 2 and 3). The
former ranges from 0.56 to 181000 mg/L (25th percentile = 834.33 mg/L, median = 1580.16 mg/L,
75th percentile = 3864.08 mg/L, and CV = 222.70%). And the latter from 0.01 to 52915 mg/L (25th
percentile = 52.87 mg/L, median = 181.27 mg/L, 75th percentile = 558.73 mg/L, and CV =
264.02%). Furthermore, the results indicate that the detectable medians of the multi-metal
concentrations (mg/L) in acid water follow the order: Fe (103.30) > Al (53.90) > Mn (8.1080) > Zn
(2.3685) > Cu (0.8010) > Ni (0.2166) > Pb (0.0700) > Cd (0.0220) > Cr (0.0200) > As (0.0108) >
Hg (0.0038), while that of the neutral/alkaline water is Mn (0.2164) > Fe (0.1700) > Zn (0.0391) >
Al (0.0350) > Cu (0.0100) > Ni (0.0050) > As (0.0034) > Cr (0.0031) > Pb (0.0012) > Cd (0.0004) >
Hg (0.0003).
*3.1.1  Contents of acid mining-affected water in China*
The multi-component concentrations of mining-affected water in both coal and metal mines
are displayed in Figs. 3 and S5, and the statistics are given in detail in Table S4. It is obvious that
the concentrations of sulfate, Fe, Mn, Al, and several HMs in the mining-affected water of most
metal mines are higher than that of coal mines, especially in mining-affected water with low pH (<
6.5). For acid mining-affected water, the pH of coal mines is approximately 1.90 - 6.50 (with a
median of 4.50), while the pH of metal mines is approximately 1.20 - 6.50 (with a median of 3.10).
The medians (IQR) of $SO_4^{2-}$ are 1381.59 (871.41, 1954.73) mg/L and 2982.00 (778.15, 10200.00)
mg/L for coal mines and metal mines, respectively. In conjunction with Fig. S6, it can be seen that
the detectable medians of multi-metal concentrations (mg/L) in coal mining-affected water are
77.41 (Fe), 12.87 (Al), 3.50 (Mn), 0.4211 (Zn), 0.1796 (Ni), 0.0431 (Cu), 0.0080 (Cr), 0.0036 (Cd),
0.0034 (As), 0.0023 (Pb), and 0.0004 (Hg), respectively. Additionally, the detectable medians of
multi-metal concentrations (mg/L) in metal mining-affected water are 152.00 (Al), 113.77 (Fe),
15.82 (Mn), 7.200 (Zn), 1.7325 (Cu), 0.2142 (Ni), 0.1498 (Pb), 0.0500 (Cr), 0.0383 (Cd), 0.0281
(As), and 0.0090 (Hg), respectively.

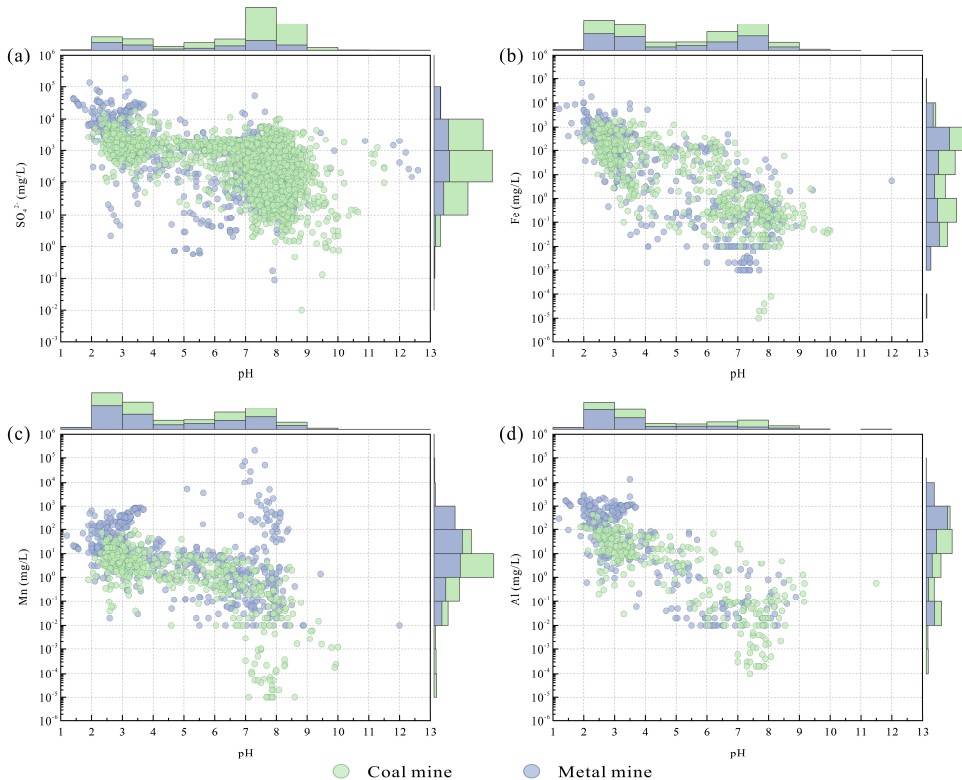


**Figure 3.** The relationship between pH and the respective concentrations including (a) $SO_4^{2-}$, (b)
Fe, (c) Mn, and (d) Al, in coal and metal mines. The binned frequency distribution of the samples
is shown along the x and y axes.

*3.1.2   Contents of non-acid mining-affected water*

Similarly, for non-acid mining-affected water, the pH value of coal mines is about 6.51 - 11.51 (with a median of 7.82), while that of metal mines is about 6.51 - 12.60 (with a median of 7.70). The medians (IQR) of $SO_4^{2-}$ are 193.51 (48.97, 582.70) mg/L and 157.41 (60.23, 425.33) mg/L for coal mines and metal mines, respectively. As shown in Fig. S6, the results indicate that the detectable medians of multi-metal concentrations (mg/L) in coal mines are in the order of Fe (0.2500) > Mn (0.0204) > Al (0.0200) > Zn (0.0048) > Ni (0.0040) > Cr (0.0022) > As (0.0016) > Cu (0.0010) > Pb (0.0003) > Hg (0.0001) > Cd (0.0000), respectively. The detectable median concentrations (mg/L) of Mn, Zn, Al, Fe, Cu, Pb, Ni, Cr, As, Cd, and Hg in metal mines are 0.7612, 0.0692, 0.0575, 0.0484, 0.0196, 0.0068, 0.0065, 0.0042, 0.0040, 0.0017, and 0.0003, respectively. In addition, the results of non-parametric tests (*i.e.*, Mann-Whitney U-test and Spearman's rank correlation) are presented in the Section S2 (Fig. S7) of the Supplement.

*3.2   Spatial patterns of mining-affected water pollution in China*

The coal mines surveyed in the study are mainly located in the northern and southwestern regions, which together account for approximately 70% of the national coal production. This localized distribution aligns closely with the pattern of coal-mining belts in China. The southwestern and southern regions of China, rich in metallic mineral resources and with complex geological conditions, have been subject to frequent or unregulated mining activities for many years. Conversely, the western and northern regions are relatively poorly endowed with metal resources (Yu et al., 2024). The mining-affected water is divided into 4 types in the study based on the multi-component characteristic, *i.e.*, with low pH, with high sulfate, with high Fe and Mn, and

with high HMs. Given that mining activities have posed great threats to the surface water and
groundwater, the classification thresholds incorporated both the distribution of the collected data
and regulatory benchmarks from the Environmental Quality Standards for Surface Water (GB
3838-2002) and the Standard for Groundwater Quality (GB/T14848-2017) in China. The categories
of water quality in the above documents are listed in Tables S5 and S6. Figs. 4 and S8 showcase
the regional patterns of the mining-affected water, and a decreasing trend in pollution levels can be
observed from South China to North China.

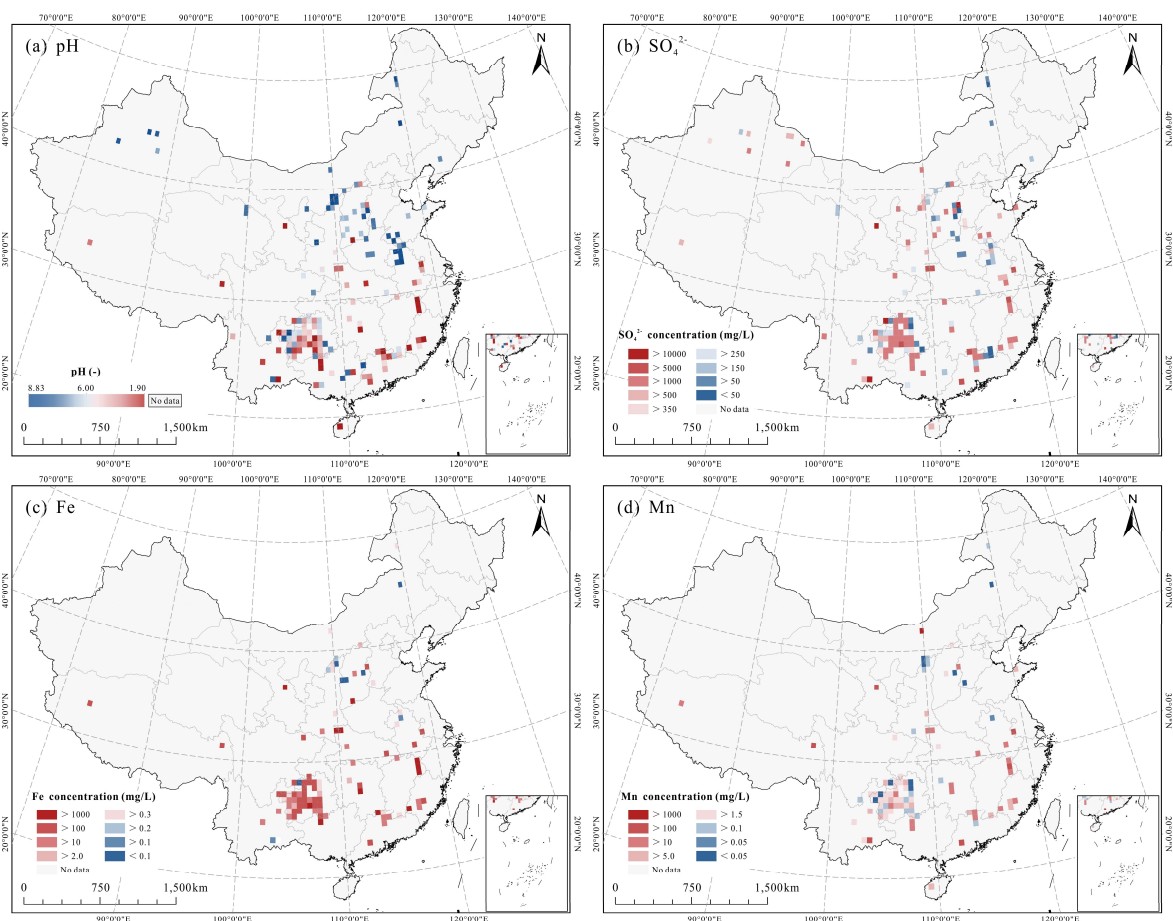


**Figure 4.** The spatial distribution of (a) pH; and the mean concentration of individual components
(mg/L) showing (b) $SO_4^{2-}$, (c) Fe, and (d) Mn, respectively, in mining-affected water on the 0.5° grid.
The classification thresholds for the main components are based on the distribution of all collected
data, as well as regulatory benchmarks from GB 3838-2002 and GB/T 14848-2017 in China.

*3.2.1 Mining-affected water with low pH*

The acid mine water is the predominant contaminant subtype, with pH values significantly below natural freshwater baselines (generally between 2.0 and 4.0). AMD is generally associated with the extraction and processing of sulfur-bearing metalliferous ore deposits (*e.g.*, pyrite, chalcopyrite, pyrrhotite, and sphalerite) and sulfide-rich coal in China (Blowes et al., 2014; Feng et al., 2014). Spatial pattern analysis revealed intense acidification hotspots (pH < 6.5) concentrated in South China, especially in the provinces of Fujian, Guangdong, Guizhou, Hubei, Hunan, Jiangxi, and Yunnan. In particular, the pH of Fuquan City in Guizhou Province reached 1.90, indicating extreme acidity. Combined with the results of the Spearman correlation analysis, a strong geochemical coupling between acidity, sulfate, and dissolved metals can be observed.

*3.2.2 Mining-affected water with high sulfate*

It is evident that there is a spatial consistency in the distribution of high-sulfate mining-affected water and acid water (Fig. 4b). The quantitative assessment of 138 data grids demonstrates that 73.9% ($n = 102$) exceed the sulfate threshold concentration of 250 mg/L. In addition, the high sulfate mining-affected water pollution is also particularly prevalent in the provinces/autonomous regions of Anhui, Hebei, Shandong, Shaanxi-Inner Mongolia, and Xinjiang, where the pH values are generally > 6.5. Contrary to typical AMD paradigms, there are two pathways to produce non-acid, high-sulfate water in the neutral/alkaline-pH systems: (i) by pyrite oxidation followed by natural neutralization, and (ii) by dissolution of sulfur-bearing and gypsum minerals. For instance, the Ordovician limestone aquifer is composed of dolomite, which is the primary source of sulfate in southwest Shandong (*e.g.*, Hongshan-Zhaili mines), Anhui (*e.g.*, Huainan-Huaibei mines) and

other mining areas. The above-mentioned spatial heterogeneities found in our study are in good
agreement with the results of Feng et al. (2014).
*3.2.3 Mining-affected water with high Fe and Mn*
Nationally, dissolved Fe and Mn concentrations far exceed the Class III threshold for
groundwater in China (GB/T14848-2017: Fe > 0.3 mg/L, Mn > 0.1 mg/L). Spatial pattern analysis
identifies six critical Fe contamination hotspots (Fig. 4c): Fujian (*e.g.*, Zijinshan copper mine),
Guangdong (*e.g.*, Lechang lead-zinc mine), Gansu (*e.g.*, Baiyin copper mine), Hunan (*e.g.*,
Shaodong coal mine), Jiangxi (*e.g.*, Dexing/Yongping copper mines) and Shannxi (*e.g.*, Baihe
pyrite mine) provinces, where the concentrations even exceed 1000 mg/L. Parallel spatial patterns
emerge for Mn contamination (Fig. 4d). Guangdong (*e.g.*, Yunfu pyrite mine), Gansu (*e.g.*, Baiyin
copper mine), Inner Mongolia (*e.g.*, Bayan Obo pyrite mine), Jiangxi (*e.g.*, Dexing copper mine),
Tibet (*e.g.*, Yulong copper mine) and Yunnan (*e.g.*, Bainiuchang polymetallic mine)
provinces/autonomous regions are the severe Mn pollution area. The hydrogeochemical cycling of
Fe and Mn in mining-affected aquatic systems is primarily governed by coupled geochemical
weathering processes and redox dynamics. Hydrodynamic conditions and water acidity play critical
roles in regulating the dissolution efficiency of these metals. It is well recognized that many metal
ores naturally contain Fe- and Mn-bearing minerals, which can release metal ions into solution
upon interaction with mine water, particularly under acidic conditions.
*3.2.4 Mining-affected water with high heavy metals*
For mining-affected waters characterized by elevated concentrations of HMs, including Cr,
Ni, Cu, Zn, As, Cd, Hg, and Pb, the identified spatial hotspots are largely consistent, mainly
distributed across the Yangtze River Basin as well as the provinces of Fujian, Gansu, Guangdong,
and Guangxi (Figs. S8). These regions are well-known as major centers for non-ferrous metal
production in China and play a critical role in the national mining industry (Zhang et al., 2016a).
Notably, in the present study, the Baiyin copper mine in Gansu Province exhibits extreme levels of
Cu, Zn, Cd, Hg, and Pb contamination. The Bainiuchang polymetallic mine, located in the
southeastern Yunnan metallogenic belt, is identified as a significant source of Cr and As. In addition,
the Zhongtiaoshan copper mining area is found to have the highest recorded concentration of Ni in
AMD, reaching 15.0 mg/L.
In connection with the results summarized in Section 3.1, the top four HMs in acid water are
Zn, Cu, Ni, and Pb, while that in non-acid water are Zn, Cu, Ni, and As. According to the review
by Yin et al. (2018), the mineral composition of copper deposits is complex and includes associated
minerals such as nickel, gold, and sulfur. Approximately 76% of the associated gold, 32.5% of the
associated silver, and 76% of the sulfur originate from copper mines in China. This reason can
explain the prevalence of Cu, Ni, and sulfate contamination hotspots in China's major copper
production bases, *i.e.*, Jiangxi (Dexing/Yongping/Dongxiang, etc.), Tongling
(Tongguanshan/Shizishan/Xinqiao, etc.), Daye (Tonglushan/Tongshankou, etc.), Zhongtiaoshan,
Baiyin, and other copper bases (Chen et al., 2013). Besides, the Zn and Pb pollution levels are
relatively higher in the Yunnan-Guizhou and Guangxi-Guangdong regions (Figs. S8d and S8h),
where are abundantly occupied by the lead-zinc ores (*e.g.*, Dachang, Daxin, Wuxu, and Fankou
lead-zinc mines, etc.). In these deposits, sphalerite (ZnS) and galena (PbS) are the principal mineral
sources of Zn and Pb, respectively (Blowes et al., 2014). With respect to As contamination in
mining-affected water, it has been demonstrated that pyrite ($FeS_2$) may contain substantial amounts
of As. Previous studies (Abraitis et al., 2004; Blanchard et al., 2007) have reported that As can
substitute for sulfur in the pyrite crystal lattice, forming As-S dianion groups. The incorporation of
As into pyrite enhances its chemical reactivity and accelerates its dissolution.
*3.3   Risks of mining-affected water in China*
Ingestion and dermal contact are the primary exposure pathways for both adults and children
residing in mining areas. The risks associated with such exposures are further exacerbated by the
persistence, mobility, and bioaccumulative potential of HMs in the environment. Among the metals
analyzed in this study, Cr, Ni, As, Cd, and Pb have been classified as carcinogenic to humans by
the International Agency for Research on Cancer. In this section, CRs are assessed specifically for
Cr, Cd, and As, as carcinogenic slope factors for the other metals are unavailable (Fig. 5).

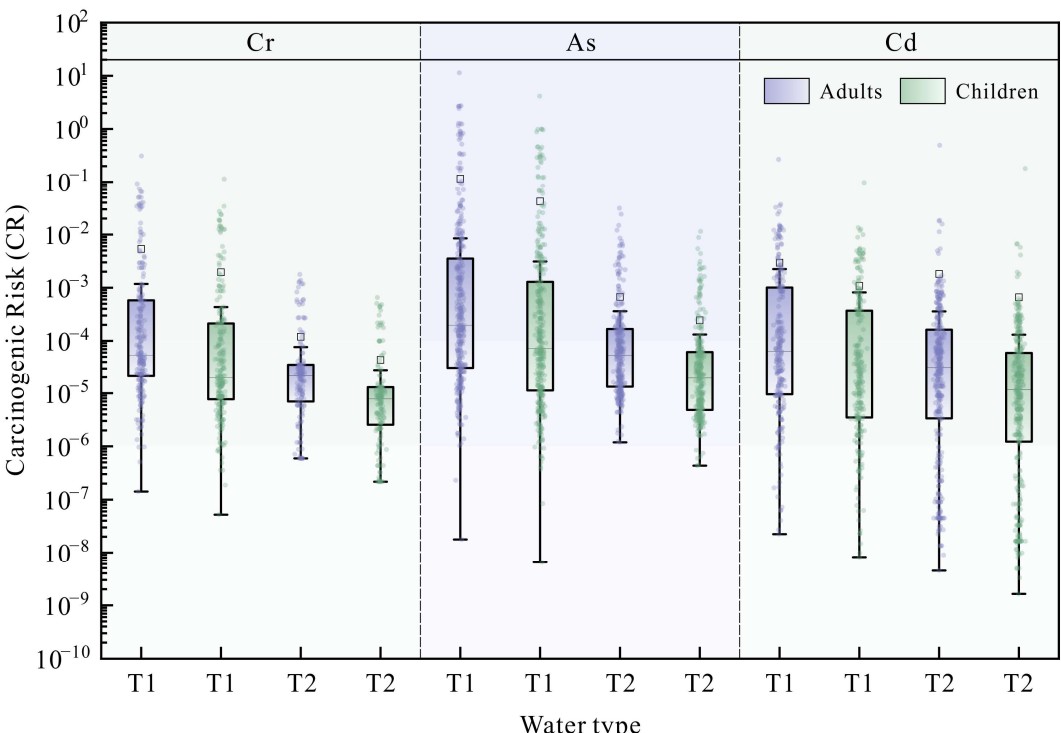


**Figure 5.** The carcinogenic risk (CR) values of Cr, As, and Cd in mining-affected water. T1
category includes mine drainage, mine water, and leachate water, while T2 category indicates
mining-affected surface water and groundwater.

Additionally, Fe, Mn, Cr, Ni, Cu, Zn, As, Cd, and Pb are taken into consideration to calculate

the cumulative NCRs for residents (Fig. 6). To better illustrate the human health risks posed by
different types of mining-affected water in China, the risk assessment is categorized into two types:
T1 and T2. The T1 category includes mine drainage, mine water, and leachate water, which pose
significant threats to the surrounding water systems. The T2 category refers to surface water and
groundwater that have been affected by mining activities.

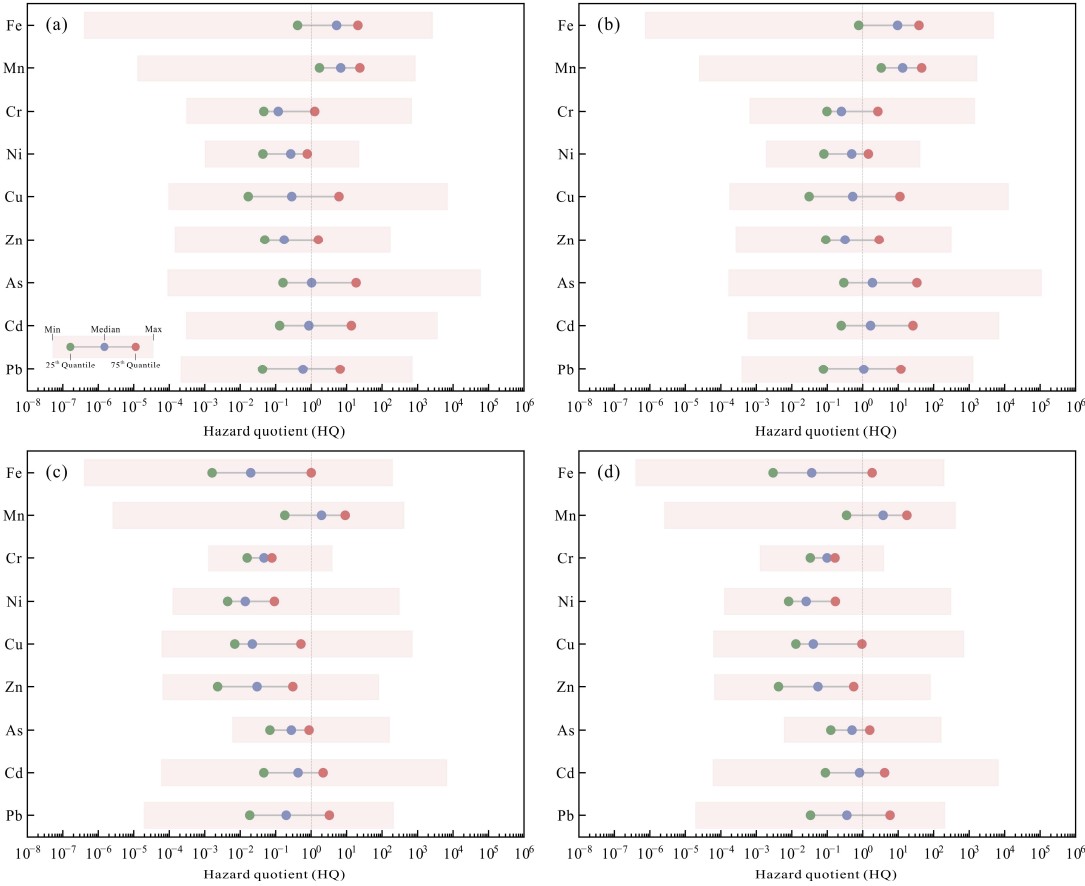


**Figure 6.** The hazard quotient (HQ) values of Fe, Mn, Cr, Ni, Cu, Zn, As, Cd, and Pb in mining-
affected water for (a) T1-Adult, (b) T1-Children, (c) T2-Adult, and (d) T2-Children, respectively.
T1 category includes mine drainage, mine water, and leachate water, while T2 category indicates
mining-affected surface water and groundwater.
*3.3.1   Carcinogenic risk of mining-affected water*

According to the guidelines established by the US EPA (2004), CR or TCR values in the range

of $10^{-6}$ to $10^{-4}$ are considered to be within the acceptable risk range. As shown in Fig. 5, the median
CR values for different population groups and water categories generally follow the order As > Cd >
Cr. This trend is consistent with the findings of Shi et al. (2018), who reported a similar risk ranking
for HMs in soils from mining areas. For T1-type water, the median CR values for all assessed
metals are below the upper limit of $10^{-4}$, except for As exposure in adults. Specifically, the median
CR values for adults are As ($1.98 \times 10^{-4}$) > Cd ($6.40 \times 10^{-5}$) > Cr ($5.39 \times 10^{-5}$), while for children
they are ($7.24 \times 10^{-5}$) > Cd ($2.34 \times 10^{-5}$) > Cr ($1.97 \times 10^{-5}$). In comparison, the CR values associated
with T2-type water are generally lower than those for T1-type water. The median values for adults
are $5.28 \times 10^{-5}$ (As), $3.14 \times 10^{-5}$ (Cd), and $2.13 \times 10^{-5}$ (Cr), while the corresponding values for
children are $1.93 \times 10^{-5}$ (As), $1.15 \times 10^{-5}$ (Cd), and $7.82 \times 10^{-6}$ (Cr). Notably, the median TCR
values for both adults and children in the mining areas assessed in this study exceed the upper
acceptable limit, reaching $3.02 \times 10^{-4}$ and $1.10 \times 10^{-4}$, respectively. In connection with the results
displayed in Fig. S9a, 68.25% of T1-type water samples and 40.27% of T2-type water samples
posed non-negligible CRs (TCR > $10^{-4}$, driven by the combined effects of Cr, Cd, and As) for adults.
The corresponding percentages for children were 51.47% (T1-type) and 23.31% (T2-type),
respectively. In terms of spatial distribution (Fig. S10), the results indicate that TCR values
associated with T1-type water exceed the acceptable threshold in 55.00% (for adults) and 40.00%
(for children) of the mining areas. For T2-type water, the proportions of mining areas with
unacceptable TCR values are 51.52% (for adults) and 29.29% (for children). These results highlight
the need for increased attention and targeted management strategies in high-risk areas.

### 3.3.2 Non-carcinogenic risk of mining-affected water

For T1-type water (Figs. 6a and 6b) with high HQ values (HQ > 1), Mn, Fe, and As are identified as the primary contributors to NCRs. The corresponding median HQ values for adults are 6.84 (Mn), 5.21 (Fe), and 1.03 (As), while the values for children are 13.26 (Mn), 9.63 (Fe), and 1.88 (As), respectively. In addition, for children, the median HQ values of Cd (1.66) and Pb (1.07) also exceed the acceptable limit of 1. In the case of T2-type water (Figs. 6c and 6d), the median HQ values for all assessed metals (except Mn) are below the USEPA threshold for both adults and children. The descending order of median HQ values is as follows: Mn (1.950 for adults, 3.752 for children) > Cd (0.424, 0.812) > As (0.274, 0.500) > Pb (0.196, 0.357) > Cr (0.047, 0.099) > Zn (0.030, 0.055) > Cu (0.022, 0.040) > Fe (0.020, 0.036) > Ni (0.014, 0.025). The results suggest that children are more sensitive to the hazardous effects of HM exposure than adults, probably due to the more conservative parameter settings used for children in the risk assessment model. In connection with the results displayed in Fig. S9b, a significant proportion of mining-affected water samples exceed NCR thresholds (HI > 1), with 88.35% of T1-type and 55.75% of T2-type samples posing potential health risks to adults. These risks primarily originated from synergistic interactions among multiple metals (Fe, Mn, Cr, Ni, Cu, Zn, As, Cd, and Pb). Children exhibit even greater vulnerability, with risk exposure percentages increasing to 91.90% (T1-type) and 63.10% (T2-type). As depicted in Fig. S11, the southern regions are predominantly identified as NCR hotspots. Specifically, for T1-type water, 89.04% (for adults) and 91.78% (for children) of mining areas exceed the acceptable HI threshold, while in the case of T2-type water, the corresponding proportions are 68.07% and 80.67%, respectively.

## 4 Discussion

### 4.1 Underlying mechanisms of pronounced mining-affected water pollution in South China

The underlying mechanisms, including climatic conditions, geological factors, and mining practices, determine the spatial patterns of mining-affected water pollution in China, especially in the highly polluted southern regions. In terms of climatic conditions, the average temperature of the coldest month is > 0°C, while that of the hottest month is > 22°C, and the annual average precipitation is generally > 1,000 mm in South China. The high temperature and precipitation create a synergistic accelerator for mine water acidification. Elevated temperatures stimulate acidophilic microbial communities (*e.g.*, *Acidithiobacillus ferrooxidans*), which enhance enzymatic activity that catalyzes sulfide mineral oxidation. Combined with high levels of precipitation, rainfall infiltrates abandoned mines, tailings ponds, and exposed ore bodies, creating a sustained water-oxygen exchange that drives sulfuric acid formation and iron oxidation processes.

In terms of geological factors, the unique geo-environmental settings of South China, characterized by rugged topography, widespread sulfur-rich strata, and high background value of metallic minerals, result in mining-affected water with high acidity and elevated concentrations of sulfate, Fe, Mn, and HMs (Sun et al., 2022). The coal-forming periods of different mines in the South China coalfields are diverse, mainly Triassic, Neoproterozoic, etc., of which the sulfur enrichment exhibits strong links to marine-land interactions. The sustained seawater intrusion-regression cycle results in elevated sulfur contents (predominantly medium and high-sulfur coals) (Ai et al., 2023; Sun et al., 2025). As illustrated in Fig. S2, the coal fields in China exhibit sulfur contents ranging from 0.02% to 10.48%, with South China's coal-bearing areas showing the highest weighted average sulfur content (2.35%), including 29.63% of high-sulfur coal. Comparatively,

those weighted average sulfur contents of coal-bearing areas in Tibet-Western Yunnan, North China,
and Northeast China are 0.94%, 0.88%, and 0.86%, respectively (Tang et al., 2015). In addition, as
shown in Fig. S3, the metal mineral resources are abundant in the southern region of China, and
the water affected by mining practices is often highly toxic, with harmful HMs such as Cd, Pb, Hg,
Cr, As, Cu, and so on, endangering the surface water and groundwater systems (Sun et al., 2022).
As to mining practices, especially those involving sulfide-bearing metalliferous ore deposits
and sulfide-rich coal mining, are intrinsically associated with AMD. Acid drainage can occur
wherever sulfide minerals are excavated and exposed to atmospheric oxygen. The main sulfide
minerals in mine wastes are pyrite ($FeS_2$) and pyrrhotite ($Fe_{1-x}S$), while other associated sulfides
are prone to oxidation and release toxic elements, including Al, As, Cd, Co, Cu, Hg, Ni, Pb and Zn,
into the water flowing through the mine tailings (Blowes et al., 2014). The oxidation of $FeS_2$ by
atmospheric oxygen can be expressed by Eqs. (8) - (11). Moreover, underground mining is the
primary exploitation method in China. Substantial mined-out areas are formed after mining
activities, inducing the accumulation of groundwater and the formation of acid mine water. In
recent years, the phenomenon has intensified because a number of mines are abandoned without
proper closure measures (Jiang et al., 2020).
$$2FeS_2 + 7O_2 + 2H_2O \rightarrow 2Fe^{2+} + 4SO_4^{2-} + 4H^+ \tag{8}$$
$$4Fe^{2+} + O_2 + 4H^+ \rightarrow 4Fe^{3+} + 2H_2O \tag{9}$$
$$Fe^{3+} + 3H_2O \rightarrow Fe(OH)_3 + 3H^+ \tag{10}$$
$$FeS_2 + 14Fe^{3+} + 8H_2O \rightarrow 15Fe^{2+} + 2SO_4^{2-} + 16H^+ \tag{11}$$
*4.2  Effects of mining-affected water pollution in China*
It is evident that acidic and metal-rich water is widespread in China, especially in the southern
areas (see Fig. 4 above and Fig. S8). As discussed in Section 4.1, the climatic conditions, geological
factors, and mining practices all play vital roles in the pronounced mining-affected water pollution
in South China. As a consequence, the contaminants pose significant risks to planetary health by
degrading surface water and groundwater quality, destroying biodiversity, and threatening human
well-being. Fig. 7 summarizes the key processes and adverse impacts of mining-affected water
pollution on the water subsystem, soil subsystem, and human health.

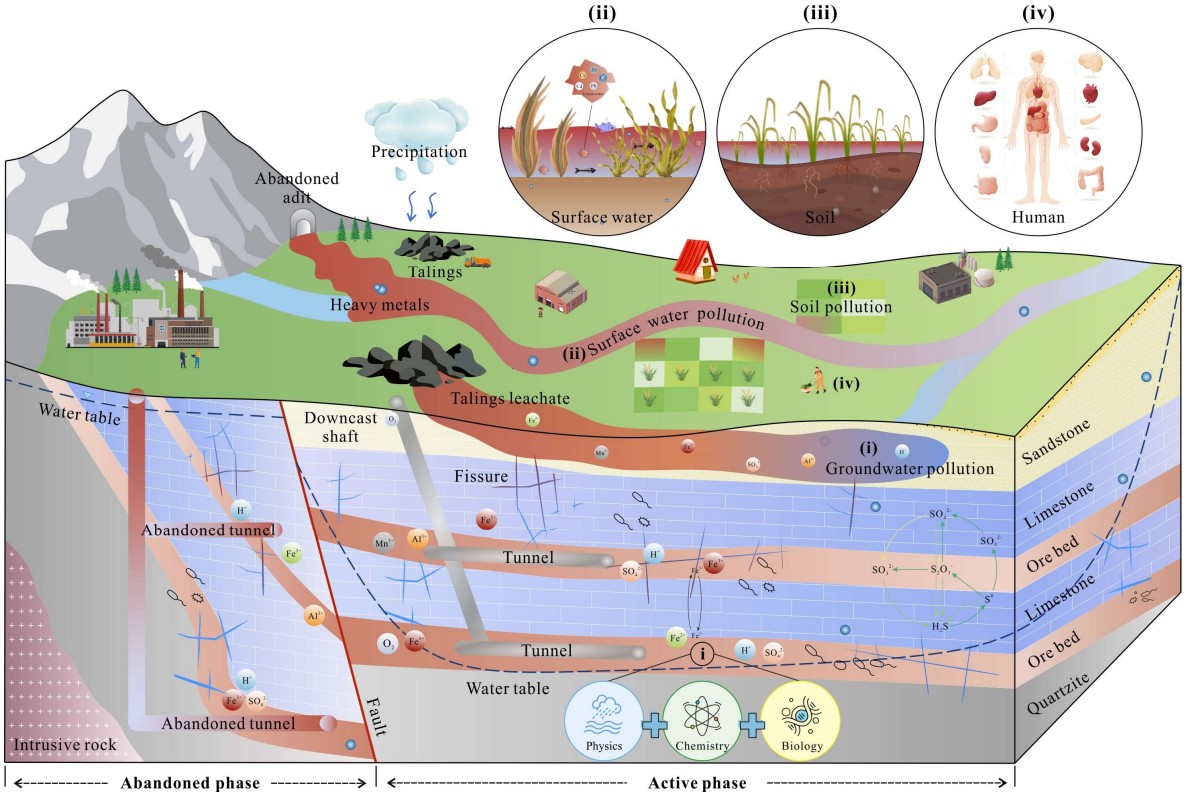


**Figure 7.** Conceptual model illustrating the pollution pathways and environmental impacts of
mining-affected water across four key subsystems: (i) groundwater, (ii) surface water, (iii) soil, and
(iv) human health, during both active mining and abandoned phases.

*Water subsystem*: As a vital component of various ecosystems, the water environment faces

increasing challenges due to the presence of diverse mining-affected water pollution (as mentioned
in Section 3.2). On the one hand, mining activities can contaminate groundwater, making it unfit
for irrigation, drinking, and other purposes. In the active mining phase, the formation of AMD is

driven by coupled physical, chemical, and biological processes initiated through sulfide mineral oxidation in the presence of oxygen and water (Fig. 7). These interconnected reactions progressively degrade groundwater quality through acidification and contaminant mobilization (Acharya and Kharel, 2020); In terms of the abandoned phase, the weathering products of exposed sulfides can serve as a source of acidity, sulfate, and dissolved metals, which may subsequently migrate and transform within the recovering groundwater (Blowes et al., 2014). On the other hand, AMD from active and abandoned mines also contaminates water bodies, lowering pH levels and destroying habitats for fish and other aquatic organisms (Ighalo et al., 2021). Toxic metals have the potential to accumulate in the food chain, especially in aquatic organisms, making them one of the most severe contaminants in surface water. Moreover, given that the metals are difficult to biodegrade, their presence has led to detrimental effects on the ecological balance of aquatic ecosystems (Gu et al., 2014; Cui et al., 2021).

**Soil subsystem**: HMs can enter the soil through mining-affected water runoff and tailings leaching, which have been increasingly detected in soil environments worldwide. Excessive HMs disrupt soil physicochemical properties, impair soil organism viability through physiological dysfunction and behavioral inhibition, and diminish agricultural productivity. Furthermore, these contaminants induce structural shifts in microbial communities, reducing both the abundance and functional diversity of keystone microbial taxa essential for biogeochemical cycling. However, the adverse effects of mining-affected water pollution on the soil subsystem are not the focus of our study. This is because Shi et al. (2023) and Yu et al. (2024) have provided a more comprehensive analysis of the pollution status, risks, and major influencing factors in coal and metal mines across China.

**Human health**: The results of the human risk assessment presented in Section 3.3 highlight that the CRs and NCRs are severe in China. Moreover, the persistence, high mobility, and bioaccumulation potential of these metals in the environment substantially increase exposure risks, thereby amplifying their adverse health effects. The eight HMs examined in this study exhibit intrinsic toxicity, posing risks to human health through bioaccumulation pathways. Mechanistically, these HMs bind to DNA strands and enzyme-active sites, inducing disruptions in cellular homeostasis, endocrine signaling, immune responses, neurophysiological functions, and reproductive-endocrine systems (Shi et al., 2023; Meng et al., 2024). For example, various injuries linked to Cr exposure include nasal irritation and ulceration, skin irritation, and perforation of the eardrum. Acute exposure to Ni can result in damage to the kidneys, liver, and brain, whereas chronic exposure can cause tissue damage. Respiratory problems, dizziness, nausea, and diarrhea are common symptoms induced by elevated Cu concentrations (Gujre et al., 2021). Zn has a significant capacity for bioaccumulation, leading to increased health risks to the immune and nervous systems via the water-food chain (Cui et al., 2021). Chronic exposure to As is associated not only with skin lesions and skin cancer, but also with neurological, respiratory, cardiovascular, and developmental effects (Zhang et al., 2024). Poisoning with Cd can cause damage to the kidneys, bones, lungs, and liver, and can even lead to cancer. (Feng et al., 2022; Liu et al., 2024). Hg can lead to serious neurological disorders in both children and adults (Rui et al., 2017). Cardiovascular, central nervous system, kidney, and fertility problems are usually associated with Pb exposure (Shi et al., 2023). Furthermore, it has recently been demonstrated that Fe is linked to pathological disorders such as Alzheimer's and Parkinson's diseases (Sahoo and Sharma, 2023).

*4.3   Implications for China's future differentiated management*

In the mining areas, the rising HMs contamination and potential health risks in surface water and groundwater call for targeted and forward-looking control strategies in China. In fact, mining regulations differ across provinces and countries, highlighting the need for site-specific frameworks and criteria. Although management may vary by location, priorities must include land use history, mine type, available technology, eco-hydrological conditions, socio-economic factors, multi-stakeholder cooperation, long-term monitoring, effective enforcement of effluent limits, and treatment standards (Acharya and Kharel, 2020). Therefore, the differentiated management in the current study is an optimized regulatory paradigm that customizes strategies to mine types (coal vs. metal) and operational status (active vs. abandoned) based on hydrogeological conditions, pollution source characteristics, and multi-system sustainability requirements. The initiative aims to implement targeted intervention and precise prevention/control to mitigate pollution risks, restore and enhance ecological functions, while concurrently safeguarding human health.

**Coal mine and metal mine**: The results imply that the water pollution status in metal mines is higher than in coal mines (Figs. 3 and S5). To some extent, policymakers should enhance their focus on regulating metal mining water contamination and devise more effective measures to reduce exposure and manage risks. The results presented in Section 3.1 imply that the characteristic contaminants in the acid water of coal mines are sulfate (with a median of 1381.59 mg/L), Fe (77.41 mg/L), and Mn (3.50 mg/L), while that of metal mines also include a variety of HMs, such as Zn (7.20 mg/L), Cu (1.73 mg/L), Ni (0.21 mg/L), Pb (0.15 mg/L) and so on. Consequently, water quality monitoring frameworks and remediation technologies should adopt site-specific strategies to account for divergent pollutant profiles in metalliferous and coal mines. These customized

approaches should integrate contaminant sources, migration pathways, and ecotoxicological impacts to ensure remediation effectiveness. Some studies have demonstrated that precipitation and neutralization are commonly used methods in coal mines, while more complex technologies, such as ion exchange or membrane separation techniques, are required to remove HMs in metal mines.

*Active mine and abandoned mine*: The differentiated management policies for active and abandoned mines aim to protect both the environment and public health across different stages of mining operations. Active mining operations require AMD prevention frameworks focusing on source control through sulfide oxidation suppression during the ore extraction and transport cycles. This requires high-frequency sensor networks for real-time contaminant flux tracking and adaptive mitigation protocols. In contrast, abandoned mine management prioritizes remediation-performance validation, integrating long-term hydrogeochemical stability monitoring with ecosystem resilience metrics. In addition, more scientific restoration strategies are critical to rebuilding the sustainable development of the water subsystem and the soil subsystem disrupted by legacy metal loads. Sustainable management also plays a pivotal role in addressing the challenges of mining-related water pollution. Emphasis should be directed to multidisciplinary partnerships and cost-effective and eco-friendly treatments, especially integrated treatment approaches that take into account the synergy of source control and end-of-pipe treatment. These elements are crucial for better understanding the complexities of mine drainage, controlling water quality degradation, and minimizing socio-economic damage.

*4.4 Reliability, limitations and prospects*

To reveal the nationwide pollution status, spatial heterogeneity, health risks, and effects of mining-affected water in China, a total of 8,433 water samples from 298 mines were integrated.

Additionally, the combination of data mining and quality assessment was employed to enhance the
reliability of the available data and build a high-quality database. However, there are still some
non-negligible limitations or uncertainties in the study.
On the one hand, the boundaries of mine sites are rarely clearly defined in the literature we
collected, which means that the spatial heterogeneity of mining-affected water pollution cannot be
accurately represented. On the other hand, the gridded data imply the southern regions, particularly
the provinces/autonomous regions of Guizhou, Guangdong, Fujian, Jiangxi, Hunan, and Guangxi,
are mining-affected water pollution hotspots. When compared with the reported sample sizes (Fig.
S4), this suggests that these areas are generally high-sampling zones, which may potentially distort
the representation of distribution. Therefore, it is of great importance to address the bias by (i)
combining the data mining and field sampling methods to investigate the potential contamination
levels in more coal and metal mines across China; (ii) balancing the sampling density within each
zone using bias correction techniques (*e.g.*, kernel density estimation and stratified spatial
resampling) to ensure the data representation; and (iii) incorporating spatial uncertainty into the
criteria to improve the spatial robustness for the assessments of mining-affected water pollution.
It is noteworthy that we cannot uncover the temporal evolution of mining-affected water
pollution due to the varying time scales of the data. Temporal variations in water chemistry (*e.g.*,
seasonal fluctuations and monsoon events) significantly impact the environmental fate of
contaminants and health risks through multiple mechanisms. During the monsoon season, heavy
rainfall flushes tailings ponds or open-pit mines, causing instantaneous spikes in HMs (*e.g.*, Cd, Cr,
and As) and sulfate concentrations. Meanwhile, the elevated groundwater levels associated with
high precipitation infiltration drive contaminant plumes along preferential pathways. These
dynamics introduce systematic biases into traditional static risk assessments. The annual or
quarterly average risk assessment model may underestimate short-term high-dose exposure risks.
Moreover, some gridded data only reflect the historical pollution status of a specific mine (*e.g.*, the
Suichang gold mine and the coal mines in the Yudong River Basin) that has undergone successful
ecological remediation and achieved good water quality levels after mining activities ceased.
Future in-depth research could focus on (i) gathering globally reported data through deep
mining and quality control and establishing a high-quality global database to better understand the
characteristics of mining-affected water pollution worldwide; (ii) identifying the key factors that
govern the transport and transformation of contaminants in surface water and groundwater systems,
during active and abandoned periods, and in coal and metal mines; (iii) enhancing the sustainable
development of coal and metal mines by AI-driven digital simulations and digital twins, which can
provide data-driven insights, optimize remediation endeavors, and advocate proactive measures to
safeguard the environment; and (iv) strengthening the studies on the synergistic measures (not only
at small-scale experimental sites but also at the mine site scale) to alleviate multifaceted
environmental challenges in the mining-affected water and achieve the development of green
mining.

## 5 Conclusions

In this study, the national status, spatial patterns, potential human health risks, and their
multifaceted implications of mining-affected water pollution have been elucidated. The new and
unique contributions of the current study are: (i) establishing a national-scale high-quality database
covering 8,433 surface water or groundwater samples (6,175 coal mine water samples and 2,258
metal mine water samples) from 298 mines (211 coal mines and 87 metal mines) in 26
provinces/autonomous regions of China; and (ii) filling the gap of the nationwide spatial patterns
of water pollution and associated health risks from both coal and metal mining activities for the
first attempt. Specifically, eight heavy metals (*i.e.*, Cr, Ni, Cu, Zn, As, Cd, Hg, and Pb) are
considered in the current study. The main results are as follows:
-    The predominant contaminants in both coal and metal mines in China are Zn, Ni and Cu. The

detectable concentrations of several heavy metals are higher in most metal mines than in coal

mines, especially in mining-affected water with low pH ($< 6.5$).

-    The order of detectable median values of water affected by coal mining is Zn (0.4211) > Ni

(0.1796) > Cu (0.0431) > Cr (0.0080) > Cd (0.0036) > As (0.0034) > Pb (0.0023) > Hg

(0.0004), while that of water affected by metal mining is Zn (7.200) > Cu (1.7325) > Ni

(0.2142) > Pb (0.1498) > Cr (0.0500) > Cd (0.0383) > As (0.0281) > Hg (0.0090).

-    The pollution hotspots and potential risks of mining-affected water (with low pH, high sulfate,

Fe, Mn, and heavy metals) are pronounced in the southern regions due to the synergistic

mechanisms of climatic conditions, geological factors, and mining practices, especially in

Guizhou, Guangdong, Fujian, Jiangxi, Hunan, and Guangxi provinces/autonomous regions.

-    The unacceptable carcinogenic risks caused by poor-quality surface water and groundwater

are observed in 51.52% (for adults) and 29.29% (for children) of the mining areas. Moreover,

severe non-carcinogenic risks are also identified in 68.07% and 80.67% of mining areas for

adults and children, respectively.

Accordingly, the findings of the study yield critical insights for designing differentiated

management measures and formulating spatially-adaptive pollution control strategies across three
key dimensions, including geographic scales (site-specific scale, provincial scale, or national scale),
mine types (coal or metal), and mining status (active or abandoned). This multidimensional
framework enables policymakers to strategically balance the trade-off between green mining
activities and human health priorities.

*Data availability*. The detailed data information can be found in Table S1.
*Author contributions*. ZYY, JS, JFW, and YY conceptualized the manuscript and its scope.
ZYY, DGL, and YYS collected the data. ZYY prepared the initial manuscript with contributions
from all co-authors. JS, JFW, YY, YYS, and JCW revised the manuscript.
*Competing interests*. The authors declare that they have no conflict of interest.
*Acknowledgements*. The authors are profoundly grateful to Editor Gabriel Rau and the three
anonymous reviewers for their insightful and constructive comments, which have greatly enhanced
the quality and clarity of the manuscript.
*Financial support*. This research is financially supported by the National Key Research and
Development Program of China (2022YFC3702200), the China National Postdoctoral Program for
Innovative Talents (BX20240165), the Jiangsu Funding Program for Excellent Postdoctoral Talents
(2024ZB125), and the Fundamental Research Funds for the Central Universities (14380228).

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
