# Peer review of "Mapping mining-affected water pollution in China: Status, patterns, risks, and"

_Hydrology and Earth System Sciences, 2024_

## Author Comment (AC1)

Response to the comments on the manuscript (HESS-2024-387) **"Mapping mining-affected water pollution in China: Status, patterns, risks, and implications"** by Ziyue Yin, Jian Song, Dianguang Liu, Jianfeng Wu[*], Yun Yang, Yuanyuan Sun, and Jichun Wu.

Note that the following text in Arial Narrow font denotes Referee's comments and in Times New Roman font denotes our response to the comments in the review. In our resubmission, the marked PDF file (HESS-2024-387_R1_marked.pdf and Supplement_R1_marked.pdf) has clearly indicated all changes to the original manuscript, tables and figures. Also, in our marked PDF file, marked in  is the text that should be removed from the original manuscript and marked in a red font is the text that has been added to the current revision. In addition, Line number(s) mentioned below can be referred to as that line numbering in the marked revised manuscript.

**Response to Referee #1's Comments**

Human activities of mining have profound impacts on water quality at local to regional scales. In this paper, the attention is paid to mapping mining-affected water pollution in China. It is achieved through the compilation of a number of publicly available surface water and groundwater datasets. In general, the paper can be valuable.

**[Response]** No change needed. We sincerely appreciate your positive and insightful comments. Hereby we have fully addressed all of your concerns into the revised manuscript and made the necessary clarifications below.

There are three major comments for further improvement of the paper:

(1) There are various types of mining activities related to the exploitation of natural resources. Underlying the activities are the spatial distribution of natural resources. For example, metals and minerals are of different distributions in China and as a result, the respective mining activities are in different places and of varying intensity. It seems that the paper does not present the big picture of spatial distributions of natural resources. Accordingly, the plot in Figure 1 is patchy, rather than comprehensive. Therefore, the authors may want to explicitly illustrate spatial distributions of natural resources as a big context of the literature survey work for the paper.

**[Response]** We appreciate your comment and have incorporated the suggestions into the revision. The belief information on natural resources in China has been presented in the revised manuscript (**Lines 95-102**) and in Section S1 of the **Supplement** (**Lines 16-57**), which serves as the cornerstone for the database development, spatial pattern analysis, and risk assessment in the study. Specifically, Figs. S1 and S2 illustrate the spatial distribution and total sulfur content of coal-bearing areas in China, and Fig. S3 exhibits the spatial distribution of the major non-ferrous mineral resources in China.

China, the second-largest economy worldwide, has various and extensive mineral resources (Li et al., 2014). It has been demonstrated that there are 171 types of mineral resources in China, with proven reserves accounting for 12% of the world's mineral resources (Hu et al., 2009). Furthermore, China is one of the largest global producers and consumers of metals and metalloids, such as Fe, Mn, Zn, Pb, Sb, and Sn (Gunson and Jian, 2001). China's coal reserves of 143,197 million tons (Mt) rank fourth globally, while its annual production of 2,971 Mt leads worldwide (Blowes et al., 2014; Ai et al., 2023).

The spatial distribution of coal fields shows significant regional differences, with dense concentrations in the coal-bearing areas of North and South China (Fig. S1). Among them, the southern Inner Mongolia, Shaanxi, Shanxi, and Henan provinces have the highest density of coal mines and mine production capacity. Besides, the coal resources of the junction of Anhui and Shandong provinces as well as Yunnan, Guizhou, Sichuan, and other provinces in southwest China are relatively rich (Xiao et al., 2021).

As shown in Fig. S3, China is rich in non-ferrous metal mineral resources. The predominant types are copper, lead-zinc, tin deposits, etc., mainly distributed in provinces such as Jiangxi, Yunnan, and Inner Mongolia. For example, the Dexing copper mine in Jiangxi province ranks as one of the largest copper deposits in China, while the Gejiu tin mine in Yunnan province is a world-renowned tin-producing area. Additionally, substantial precious metal mineral resources (gold and silver deposits) are predominantly located in Shandong, Henan, and Guizhou provinces. For example, the Zhaoyuan gold mine in Shandong province is a historically significant gold-producing region.

It is noteworthy that the national mineral deposit database of China developed by Li et al. (2019), covering 232 mineral resources in 27,569 deposits in 29 provinces (cities or districts), is of great importance to study the national natural resources. It can help readers catch more authoritative information, such as ore species, deposit name, location, latitude (N), longitude (E), genetic type of deposit, paragenetic mineral, associated mineral, deposit scale, ore-forming age, and mining status, enabling comprehensive analysis of China's natural resources.

Ai Y.L., Chen, H.P., Chen, M.F., Huang, Y., Han, Z.T., Liu, G., Gao, X.B., Yang, L.H., Zhang, W.Y., Jia, Y.F., and Li, J.: Characteristics and treatment technologies for acid mine drainage from abandoned coal mines in major coal-producing countries. J. China Coal Soc., 48(12), 4521-4535 (in Chinese with English abstract), https://doi.org/10.13225/j.cnki.jccs.2022.1846, 2023.

Blowes, D.W., Ptacek, C.J., Jambor, J.L., Weisener, C.G., Paktunc, D., Gould W.D., and Johnson, D.B. The geochemistry of acid mine drainage. Treatise on Geochemistry (Second Edition), 11, 131-190, https://doi.org/10.1016/B978-0-08-095975-7.00905-0, 2014.

Gunson, A.J. and Jian, Y.: Artisanal mining in the People's Republic of China. International Institute of Environment and Development, 2001.

Hu, R.Z., Liu, J.M., and Zhai, M.G.: Mineral resources science in China: a roadmap to 2050. Science Press, Beijing, 2009.

Li, C.Y., Liu, F.Y., Li, J., He, C.Z., Wang, X.C., and Wang F.: National mineral deposit database of China. Geology in China, 46(S2), 1-8 (in Chinese with English abstract), https://doi.org/10.12029/gc2019Z201, 2019.

Li, Z., Ma, Z., van der Kuijp, T.J., Yuan, Z., and Huang, L.: A review of soil heavy metal pollution from mines in China: Pollution and health risk assessment. Sci. Total Environ., 468-469, 843-853, https://doi.org/10.1016/j.scitotenv.2013.08.090, 2014.

Xiao, W., Chen, W.Q., and Deng, X.Y.: Coupling and coordination of coal mining intensity and social-ecological resilience in China. Ecol. Indic., 131, 108167, https://doi.org/10.1016/j.ecolind.2021.108167, 2021.

(2) It is known that the mining of coal has a considerable impact on water quality, especially in West China. On the other hand, there is a lack of in-depth investigation of coal mining. Given the importance of coal mining and the existence of extensive studies, the authors may want to present a detailed analysis.

References:

Yajun, S.U.N., Ge, C.H.E.N., Zhimin, X.U., Huiqing, Y., Yuzhuo, Z., Lijie, Z., Xin, W., Chenghang, Z. and Jieming, Z., 2020. Research progress of water environment, treatment and utilization in coal mining areas of China. Journal of China Coal Society, 45(1), pp.304-316.

Zhang, X., Li, X. and Gao, X., 2016. Hydrochemistry and coal mining activity induced karst water quality degradation in the Niangziguan karst water system, China. Environmental Science and Pollution Research, 23, pp.6286-6299.

Qu, S., Liang, X., Liao, F., Mao, H., Xiao, B., Duan, L., Shi, Z., Wang, G. and Yu, R., 2023. Geochemical fingerprint and spatial pattern of mine water quality in the Shaanxi-Inner Mongolia Coal Mine Base, Northwest China. Science of The Total Environment, 854, p.158812.

[Response] Thank you for your constructive suggestions. An in-depth analysis of the considerable impacts of coal mining activities on water quality has been added to the revised manuscript based

on the results obtained from the following references (**Lines 100-105**, **Lines 551-555**, and **Lines 559-564**):

Coal is the predominant energy source for both domestic and industrial use in China, with reserves of 143,197 million tons (Mt) ranking fourth globally, while its annual production of 2,971 Mt leads worldwide (Blowes et al., 2014; Ai et al., 2023). However, coal extraction inevitably generates substantial amounts of mine water, resulting in a series of water environmental issues (Zhang et al., 2016; Qu et al., 2023). Current estimate shows a 2:1 mine water to coal production ratio, with approximately 2 tons of mine water produced per ton of extracted coal in China (Gu et al., 2021). Coal mining activities, especially those involving sulfide-rich coal mining, are intrinsically associated with acid mine drainage (AMD). Our investigation in this study focuses on mining regions in northern and southwestern China, which account for approximately 70% of the national coal production. The spatial hotspots of mining-affected water pollution are mainly distributed in the southern regions, especially in Guizhou, Guangdong, Fujian, Jiangxi, Hunan, and Guangxi provinces/autonomous regions. As illustrated in Fig. S2, the coal fields in China exhibit sulfur contents ranging from 0.02% to 10.48%, with South China's coal-bearing areas showing the highest weighted average sulfur content (2.35%), including 29.63% of high-sulfur coal. Comparatively, those weighted average sulfur contents of coal-bearing areas in Tibet-Western Yunnan, North China, and Northeast China are 0.94%, 0.88%, and 0.86%, respectively (Tang et al., 2015). The main sulfide minerals in mine wastes are pyrite ($FeS_2$) and pyrrhotite ($Fe_{1-x}S$), while other associated sulfides are prone to oxidation and release toxic elements, including Al, As, Cd, Co, Cu, Hg, Ni, Pb and Zn, into the water flowing through the mine tailings (Blowes et al., 2014). Consequently, mining-affected water is characterized by the presence of diverse contaminants, including excessive sulfate, fluoride, and toxic heavy metals (Sun et al., 2020; 2025).

Gu, D.Z., Li, J.F., Cao, Z.G., Wu, B.Y., Jiang, B.B., Yang, Y., Yang, J., Chen, Y.P.: Technology and engineering development strategy of water protection and utilization of coal mine in China. J. China Coal Soc., 46(10), 3079-3089 (in Chinese with English abstract), https://doi.org/10.13225/j.cnki.jccs.2021.0917, 2021.

Qu, S., Liang, X., Liao, F., Mao, H., Xiao, B., Duan, L., Shi, Z., Wang, G. and Yu, R.: Geochemical fingerprint and spatial pattern of mine water quality in the Shaanxi-Inner Mongolia Coal Mine Base, Northwest China. Sci. Total Environ., 854, 158812, https://doi.org/10.1016/j.scitotenv.2022.158812, 2023.

Sun, Y.J., Chen, G., Xu, Z.M., Yuan, H.Q., Zhang, Y.Z., Zhou, L.J., Wang, X., Zhang, C.H., and Zheng, J.M.: Research progress of water environment, treatment and utilization in coal mining areas of China. J. China Coal Soc., 45(1), 304-316 (in Chinese with English abstract), https://doi.org/10.13225/j.cnki.jccs.YG19.1654, 2020.

Sun, Y.J., Guo, J., Xu, Z.M., Zhang, L., Chen, G., Xiong, X.F., Hua, J.F., Mu, L.J., and Wu, W.X.: Spatial distribution characteristics of mine water quality in coal mining areas of China and technological approaches for mine water

treatment. J. China Coal Soc., 50(1), 584-599 (in Chinese with English abstract), https://doi.org/10.13225/j.cnki.jccs.YG24.1547, 2025.

Tang, Y.G., He, X., Cheng, A.G., Li, W.W., Deng, X.J., Wei, Q., and Li, L.: Occurrence and sedimentary control of sulfur in coals of China. J. China Coal Soc., 40(9), 1977-1988 (in Chinese with English abstract), https://doi.org/10.13225/j.cnki.jccs.2015.0434, 2015.

Zhang, X., Li, X., and Gao, X.: Hydrochemistry and coal mining activity induced karst water quality degradation in the Niangziguan karst water system, China. Environ. Sci. Pollut. Res., 23, 6286-6299, https://doi.org/10.1007/s11356-015-5838-z, 2016.

(3) There are serious concerns on heavy metal pollution in recent years. Previously, there have been a few review papers. What new insights (findings) does this paper make?

References:

Cheng, S., 2003. Heavy metal pollution in China: origin, pattern and control. Environmental science and pollution research, 10, pp.192-198.

He, B., Yun, Z., Shi, J. and Jiang, G., 2013. Research progress of heavy metal pollution in China: Sources, analytical methods, status, and toxicity. Chinese Science Bulletin, 58, pp.134-140.

Hu, H., Jin, Q. and Kavan, P., 2014. A study of heavy metal pollution in China: Current status, pollution-control policies and countermeasures. Sustainability, 6(9), pp.5820-5838.

**[Response]** The point is well taken. Based on your suggestions, we have reviewed the following references and cited them in the revised text and reference lists. Indeed, more than two decades ago, Cheng (2003) reviewed the sources of heavy metal pollution (*i.e.*, industrial emissions, wastewater irrigation, and waste fertilization) in the case sites, analyzed heavy metals in food and their transfer through the food web, and provided internationally available pollution control measures. About 10 years later, He et al. (2013) reviewed the status, sources, toxicity and potential risks, and possible reduction strategies of heavy metal pollution (especially for Pb, Hg, Cd, Cr, and As) in China. Also, Hu et al. (2014) reviewed the sources of heavy metal pollution (*i.e.*, waste gas, wastewater, and solid waste), discussed the policies (*e.g.*, the 12th Five-Year Plan on Prevention and Control of Heavy Metal Pollution) and challenges of controlling heavy metal pollution, and proposed corresponding countermeasures to mitigate heavy metal pollution by increasing the green GDP, reducing the heavy metals in fuel, using more renewable energy, and adopting market-based approaches.

Undoubtedly, the previous studies mentioned above have provided a solid basis for exploring the issues of heavy metal pollution originating from multiple pollution sources. In comparison to the previous studies, our study focuses on mapping mining-affected water pollution in China,

elaborating on its status, patterns, risks, and implications. The new and unique contributions of the current study are: (i) establishing a national-scale high-quality database covering 8,433 surface water or groundwater samples (6,175 coal mine water samples and 2,258 metal mine water samples) from 298 mines (211 coal mines and 87 metal mines) in 26 provinces/autonomous regions of China; and (ii) filling the gap of the nationwide spatial patterns of water pollution and associated health risks from both coal and metal mining activities for the first attempt. Specifically, eight heavy metals (*i.e.*, Cr, Ni, Cu, Zn, As, Cd, Hg, and Pb) are considered in the current study based on a national-scale high-quality hydrochemical database. The new results show that Zn, Ni, and Cu are the predominant contaminations contaminants of both coal and metal mines in China. The detectable concentrations of several heavy metals are higher in most metal mines than in coal mines, especially in mining-affected water with low pH ($< 6.5$). The order of detectable median values of water affected by coal mining is Zn (0.4211) > Ni (0.1796) > Cu (0.0431) > Cr (0.0080) > Cd (0.0036) > As (0.0034) > Pb (0.0023) > Hg (0.0004), while that of water affected by metal mining is Zn (7.200) > Cu (1.7325) > Ni (0.2142) > Pb (0.1498) > Cr (0.0500) > Cd (0.0383) > As (0.0281) > Hg (0.0090). In terms of spatial patterns, the pollution hotspots and potential risks of mining-affected water (with low pH, high sulfate, Fe, Mn, and heavy metals) are pronounced in the southern regions, especially in Guizhou, Guangdong, Fujian, Jiangxi, Hunan, and Guangxi provinces/autonomous regions. These phenomena are closely linked to the underlying mechanisms, such as climatic conditions, geological factors, and mining practices. Accordingly, the findings of the study yield critical insights for designing differentiated management measures and formulating spatially-adaptive pollution control strategies across three key dimensions, including geographic scales (site-specific scale, provincial scale, or national scale), mine types (coal or metal), and mining status (active or abandoned). This multidimensional framework enables policymakers to strategically balance the trade-off between green mining activities and human health priorities (**Lines 743-771**).

Cheng, S.: Heavy metal pollution in China: origin, pattern and control. Environ. Sci. Pollut. Res., 10(3), 192-198. https://doi.org/10.1065/espr2002.11.141.1, 2003.

He, B., Yun, Z.J., Shi, J.B., and Jiang, G.B.: Research progress of heavy metal pollution in China: Sources, analytical methods, status, and toxicity. Chin. Sci. Bull., 58(2), 134-140, https://doi.org/10.1007/s11434-012-5541-0, 2013.

Hu, H., Jin, Q., and Kavan, P.: A study of heavy metal pollution in China: Current status, pollution-control policies and countermeasures. Sustainability, 6, 5820-5838, https://doi.org/10.3390/su6095820, 2014.

---

## Author Comment (AC2)

Response to the comments on the manuscript (HESS-2024-387) **"Mapping mining-affected water pollution in China: Status, patterns, risks, and implications"** by Ziyue Yin, Jian Song, Dianguang Liu, Jianfeng Wu[*], Yun Yang, Yuanyuan Sun, and Jichun Wu.

Note that the following text in Arial Narrow font denotes Referee's comments and in Times New Roman font denotes our response to the comments in the review. In our resubmission, the marked PDF file (HESS-2024-387_R1_marked.pdf and Supplement_R1_marked.pdf) has clearly indicated all changes to the original manuscript, tables and figures. Also, in our marked PDF file, marked in a green strikethrough font is the text that should be removed from the original manuscript and marked in a red font is the text that has been added to the current revision. In addition, Line number(s) mentioned below can be referred to as that line numbering in the marked revised manuscript.

**Response to Referee #2's Comments**

The manuscript aims to provide a comprehensive assessment of mining-affected water pollution across China by compiling a large dataset (8433 water samples from 298 mines). The study evaluates spatial patterns, assesses both carcinogenic and non-carcinogenic risks to human health, and discusses management implications for both coal and metal mining areas. While the work is well supported by extensive data and robust methodologies, questions remain regarding the novelty of the contribution, as the manuscript does not clearly delineate how its findings significantly extend beyond previous studies.

**[Response]** We sincerely thank you for your constructive and conscientious suggestions. Hereby we have fully incorporated and addressed all the comments in the revised manuscript and given a point-by-point response as below. In particular, a more explicit statement of our novel contributions relative to the existing literature (*e.g.*, Cheng, 2003; He et al., 2013; Hu et al., 2013; Feng et al., 2014; Sun et al., 2025) has been added to the **Conclusion** (**Lines 743-771**) to address your concern that "the manuscript does not clearly delineate how its findings significantly extend beyond previous studies".

It is noteworthy that previous studies predominantly concentrated on localized water pollution from individual coal or metal mines, while national-scale assessments have primarily addressed impacts exclusively attributed to coal mining activities. The new and unique contributions of the

current study are: (i) establishing a national-scale high-quality database covering 8,433 surface water or groundwater samples (6,175 coal mine water samples and 2,258 metal mine water samples) from 298 mines (211 coal mines and 87 metal mines) in 26 provinces/autonomous regions of China; and (ii) filling the gap of the nationwide spatial patterns of water pollution and associated health risks from both coal and metal mining activities for the first attempt. Specifically, eight heavy metals (i.e., Cr, Ni, Cu, Zn, As, Cd, Hg, and Pb) are considered in the current study based on a national-scale high-quality hydrochemical database. The new results show that Zn, Ni, and Cu are the predominant contaminations contaminants of both coal and metal mines in China. The detectable concentrations of several heavy metals are higher in most metal mines than in coal mines, especially in mining-affected water with low pH ($< 6.5$). The order of detectable median values of water affected by coal mining is Zn (0.4211) > Ni (0.1796) > Cu (0.0431) > Cr (0.0080) > Cd (0.0036) > As (0.0034) > Pb (0.0023) > Hg (0.0004), while that of water affected by metal mining is Zn (7.200) > Cu (1.7325) > Ni (0.2142) > Pb (0.1498) > Cr (0.0500) > Cd (0.0383) > As (0.0281) > Hg (0.0090). In terms of spatial patterns, the pollution hotspots and potential risks of mining-affected water (with low pH, high sulfate, Fe, Mn, and heavy metals) are pronounced in the southern regions, especially in Guizhou, Guangdong, Fujian, Jiangxi, Hunan, and Guangxi provinces/autonomous regions. These phenomena are closely linked to the underlying mechanisms, such as climatic conditions, geological factors, and mining practices. Accordingly, the findings of the study yield critical insights for designing differentiated management measures and formulating spatially-adaptive pollution control strategies across three key dimensions, including geographic scales (site-specific scale, provincial scale, or national scale), mine types (coal or metal), and mining status (active or abandoned). This multidimensional framework enables policymakers to strategically balance the trade-off between green mining activities and human health priorities.

Cheng, S.: Heavy metal pollution in China: origin, pattern and control. Environ. Sci. Pollut. Res., 10(3), 192-198. https://doi.org/10.1065/espr2002.11.141.1, 2003.

Feng, Q., Li, T., Qian, B., Zhou, L., Gao, B., and Yuan, T.: Chemical Characteristics and Utilization of Coal Mine Drainage in China. Mine Water Environ., 33, 276-286, https://doi.org/10.1007/s10230-014-0271-y, 2014.

He, B., Yun, Z.J., Shi, J.B., and Jiang, G.B.: Research progress of heavy metal pollution in China: Sources, analytical methods, status, and toxicity. Chin. Sci. Bull., 58(2), 134-140, https://doi.org/10.1007/s11434-012-5541-0, 2013.

Hu, H., Jin, Q., and Kavan, P.: A study of heavy metal pollution in China: Current status, pollution-control policies and countermeasures. Sustainability, 6, 5820-5838, https://doi.org/10.3390/su6095820, 2014.

Sun, Y.J., Guo, J., Xu, Z.M., Zhang, L., Chen, G., Xiong, X.F., Hua, J.F., Mu, L.J., and Wu, W.X.: Spatial distribution characteristics of mine water quality in coal mining areas of China and technological approaches for mine water

treatment. J. China Coal Soc., 50(1), 584-599 (in Chinese with English abstract), https://doi.org/10.13225/j.cnki.jccs.YG24.1547, 2025.

**Novelty and Original Contribution**

Strengths: The study assembles a large national dataset and applies risk assessment models to evaluate health impacts, which is commendable. The spatial mapping of contamination hotspots and risk distribution provides valuable insights for policy-making.

**[Response]** No change needed. Thank you for your positive comments.

Concerns: One key issue is that the manuscript does not sufficiently highlight what is new compared to earlier studies. Although the scale of the data collection is impressive, the paper lacks a clear statement of its novel contributions relative to existing literature. The authors could enhance the manuscript by emphasizing unique aspects—such as new methodological approaches, previously unreported spatial trends, or innovative risk assessment strategies—that set this work apart.

**[Response]** The point is well taken. We have incorporated your concerns into the revision and a more explicit statement has been added to the revised **Conclusion**, highlighting the novelty and practical implications of our manuscript relative to the existing literature (**Lines 743-771**). See also the response to your general comment above.

**Methodological Rigor and Data Quality**

Strengths: The methodology is generally robust, with clear criteria for data quality control and appropriate use of standard risk assessment models (e.g., those provided by the US EPA). The division of water samples (acidic vs. neutral/alkaline) and the differentiation between coal and metal mines are well executed.

**[Response]** No change needed. Thank you for your positive comments.

Suggestions: To further strengthen the paper, the authors should elaborate on how potential biases (e.g., variations in sample density among regions) were addressed. Additionally, more detailed statistical tests comparing water quality parameters between different mining types (such as using non-parametric tests) could provide further evidence for the observed differences.

**[Response]** Comment accepted. Indeed, there are potential biases caused by variations in sample density among regions. Therefore, our future in-depth research will attempt to address the biases by (i) combining the data mining and field sampling methods to investigate the potential contamination levels in more coal and metal mines across China; (ii) balancing the sampling

density within each zone using bias correction techniques (*e.g.*, kernel density estimation and stratified spatial resampling) to ensure the data representation; and (iii) incorporating spatial uncertainty into the criteria to improve the spatial robustness for the assessments of mining-affected water pollution (**Lines 710-716**). Furthermore, we have added the results of non-parametric tests (*i.e.*, Mann-Whitney U-test and Spearman's rank correlation) to further support the differences observed in our study based on your suggestions (**Lines 117-131** in the **Supplement**):

Non-parametric tests do not rely on assumptions about the distribution of the data and are suitable for non-normally distributed datasets or those containing outliers (Cardew, 2003). These methods statistically compare central tendencies, typically represented by medians, rather than means. The result of the Mann-Whitney U-test ($p < 0.05$) shows a statistically significant difference in the critical parameters (except Fe) of mining-affected water based on the different mine types (coal mine vs. metal mine), indicating the differences caused by geological factors, mining practices, surrounding environment, etc. Besides, Fig. S7 shows the Spearman correlation coefficients between the hydrochemical compositions in the mining-affected water. It can be seen that strong negative correlations are observed between pH and $SO_4^{2-}$, Fe, Mn, Al, and heavy metals while positive correlations are observed between $SO_4^{2-}$ and metal components, implying that the spatial consistency of acid water, high sulfate, high Fe and Mn, and high heavy metal mining-affected water.

Cardew, P.T.: A method for assessing the effect of water quality changes on plumbosolvency using random daytime sampling. Water Res., 37(12), 2821-2832, https://doi.org/10.1016/S0043-1354(03)00120-9, 2003.

**Presentation and Interpretation of Results**

Strengths: The results are logically presented, starting from the basic water quality parameters, moving on to spatial distribution patterns, and culminating in detailed risk assessments for different populations. Figures (e.g., maps and boxplots) support the textual description and help visualize the trends effectively.

**[Response]** No change needed. Thank you for your positive comments.

Suggestions: Although the numerical details are extensive, the manuscript may benefit from a more concise presentation. For example, summarizing key quantitative findings in a table could improve clarity. Additionally, while the spatial patterns are well described, a deeper discussion on the underlying geochemical or environmental processes that cause these trends would better contextualize the results.

**[Response]** Thank you for your insightful comments. To improve the clarity of the manuscript, key quantitative results (*e.g.*, statistics of critical parameters for acid and neutral/alkaline water

across different mines) have been summarized in Table 1 in the revised manuscript and Table S4 in the **Supplement**. Moreover, a further elaboration of the underlying mechanisms (*e.g.*, geochemical conditions and environmental processes) driving the spatial patterns of mining-affected water pollution in China, especially in the highly polluted southern regions, has been added in Section 4.1 of the revised manuscript (**Lines 534-573**).

**Structure and Coherence of the Argument**

Strengths: The manuscript follows a conventional structure (introduction, methodology, results, discussion, conclusion) that makes it easy to follow. The discussion ties the findings back to the broader context of water pollution management.

[**Response**] No change needed. Thank you for your positive comments.

Suggestions: The transition between sections—especially from the results to the discussion—could be smoother. Explicitly linking how each result addresses the stated objectives would reinforce the coherence of the argument. Also, highlighting the novelty and practical implications of the work in the conclusion would help reinforce the manuscript's contribution.

[**Response**] Comment accepted. We have rewritten the transition between sections to reinforce the coherence of the argument. Moreover, a detailed discussion of the novelty and practical implications of our manuscript has been supplemented in the **Conclusion** to highlight the unique contribution and valuable addition to the field of environmental hydrology (**Lines 743-771**).

**Figures, Tables, and Visual Aids**

Strengths: Visual aids are generally clear and provide a good overview of the data distribution and risk maps. The integration of detailed figures (such as spatial distribution maps and risk assessment graphs) adds significant value to the manuscript.

[**Response**] No change needed. Thank you for your positive comments.

Suggestions: Ensure that all figures have clear legends and consistent formatting. It might be beneficial to include a summary table that aggregates the key findings (e.g., median values of critical parameters across different mine types) to enhance readability.

[**Response**] Thank you for your constructive suggestions. We have reviewed/revised all figures to ensure that they have clear legends and consistent formatting. To improve overall readability, the

summary tables (*i.e.*, Table 1 in the revised manuscript and Table S4 in the **Supplement**) showing the statistics of critical parameters for different mine types have been added to the current revision.

**Language and Style**

Strengths: The manuscript is written in clear, professional English with an appropriate academic tone. Technical terms are defined upon first use, and the text is generally free of major grammatical errors.

[**Response**] No change needed. Thank you for your positive comments.

Suggestions: A few sentences could be simplified to improve readability. In particular, some complex sentences in the introduction and discussion might be broken into shorter, more digestible statements. Maintaining consistency in terminology (for instance, ensuring that terms like "differentiated management" are clearly defined) will also help in reinforcing the manuscript's clarity.

[**Response**] Comment accepted. We have simplified some complex sentences in the **Introduction** and **Discussion** to improve the readability of the manuscript. Moreover, a clear definition of terms like "differentiated management" has been added in the revised manuscript to reinforce the manuscript's clarity (**Lines 650-655**):

The differentiated management mentioned in the current study is an optimized regulatory paradigm that customizes strategies to mine types (coal vs. metal) and operational status (active vs. abandoned) based on hydrogeological conditions, pollution source characteristics, and multi-system sustainability requirements. The initiative aims to implement targeted intervention and precise prevention/control to mitigate pollution risks, restore and enhance ecological functions, while concurrently safeguarding human health.

**Conclusion**

The manuscript presents an extensive dataset and a rigorous analysis of mining-affected water pollution in China, offering useful insights for environmental management and policy-making. However, the work would benefit from a more explicit discussion of its novelty compared to previous studies. Clarifying and emphasizing the unique contributions—whether in data scale, methodological advancements, or new insights into spatial and health risk patterns—would significantly strengthen the paper. With these revisions, the manuscript could represent a valuable addition to the field of environmental hydrology.

[**Response**] We sincerely appreciate your conscientious and constructive comments. A more explicit discussion of the novelty compared to previous studies has been added to the revised manuscript (**Lines 743-771**), to provide new insights into the spatial patterns and health risks of mining-affected water pollution at the national scale, and to clarify and emphasize the unique

contributions of our study. We believe that your insightful comments on 'Novelty and Original Contribution', 'Methodological Rigor and Data Quality', 'Presentation and Interpretation of Results', 'Structure and Coherence of the Argument', 'Figures, Tables, and Visual Aids', and 'Language and Style' have led to significant improvements of the revised manuscript.

---

## Author Comment (AC3)

Response to the comments on the manuscript (HESS-2024-387) **"Mapping mining-affected water pollution in China: Status, patterns, risks, and implications"** by Ziyue Yin, Jian Song, Dianguang Liu, Jianfeng Wu[*], Yun Yang, Yuanyuan Sun, and Jichun Wu.

Note that the following text in Arial Narrow font denotes Referee's comments and in Times New Roman font denotes our response to the comments in the review. In our resubmission, the marked PDF file (HESS-2024-387_R1_marked.pdf and Supplement_R1_marked.pdf) has clearly indicated all changes to the original manuscript, tables and figures. Also, in our marked PDF file, marked in a green strikethrough font is the text that should be removed from the original manuscript and marked in a red font is the text that has been added to the current revision. In addition, Line number(s) mentioned below can be referred to as that line numbering in the marked revised manuscript.

**Response to Referee #3's Comments**

The study delivers a thorough and spatially explicit examination of mining-induced water pollution and associated health risks across China, utilizing an extensive dataset comprising 8,433 samples. The differentiation between coal and metal mines, along with the identification of southern China as a pollution hotspot, provides valuable insights for region-specific policy formulation. Below are several constructive suggestions for refining the manuscript:

**[Response]** We sincerely appreciate your constructive comments and the recommendation for minor revisions of the manuscript. Moreover, we have made the necessary changes to the original manuscript and hereinafter provided a point-by-point response.

1. While the spatial heterogeneity of pollution is convincingly presented, the underlying mechanisms driving the pronounced contamination in southern China (e.g., geological factors, mining practices, or climatic conditions) warrant further elaboration. Incorporating a brief discussion that connects regional geochemistry or historical mining activities to observed pollution patterns would enhance the robustness of the analysis.

**[Response]** Thank you for your positive and constructive suggestions. As you suggested, a further elaboration of the underlying mechanisms (*e.g.*, geological factors, mining practices, and climatic conditions) driving the pronounced contamination in South China has been added in Section 4.1 of

the revised manuscript. Furthermore, to enhance the robustness of the analysis, the potential relation between the observed water pollution patterns and both regional geochemical characteristics and historical mining activities has been discussed in detail (**Lines 534-573**):

[revised manuscript text omitted]

Tang, Y.G., He, X., Cheng, A.G., Li, W.W., Deng, X.J., Wei, Q., and Li, L.: Occurrence and sedimentary control of sulfur in coals of China. J. China Coal Soc., 40(9), 1977-1988 (in Chinese with English abstract), https://doi.org/10.13225/j.cnki.jccs.2015.0434, 2015.

2. The health risk assessment (e.g., 51.52% carcinogenic risk for adults) raises significant concerns but lacks sufficient methodological detail. Please specify the exposure parameters employed (e.g., ingestion rates,

body weight assumptions) and the toxicity thresholds applied. Additionally, clarify whether the identified risks stem from specific contaminants (e.g., arsenic, cadmium) or synergistic interactions among multiple pollutants.

[Response] Comment accepted. The main parameters used for the human health risk assessment (*e.g.*, ingestion rates, exposure frequency, body weight, etc.) are presented in Table S2. The values of permeability coefficient of skin ($K_p$), reference dose ($RfD_o$), gastrointestinal digestion coefficient ($ABS_{GI}$), and slope factor (SF) for each element are described in Table S3 in Section S3 of the **Supplement**. In the study, Cr, Cd, and As are considered for the calculation of the carcinogenic risks. On the other hand, Fe, Mn, Cr, Ni, Cu, Zn, As, Cd, and Pb are taken into consideration to identify the non-carcinogenic risks. Note that the identified risks (*e.g.*, 51.52% carcinogenic risk for adults) stem from synergistic interactions among multiple pollutants, which have been clarified in the revised manuscript to avoid unnecessary misunderstandings (**Lines 479-482** and **Lines 517-520**).

3. The temporal dimension of water sampling remains ambiguous. Were the samples collected across different seasons or years? Temporal variability in water chemistry (e.g., the influence of monsoon events on metal mobility) could significantly affect risk estimates and merits further exploration in the discussion section.

[Response] We appreciate your insight. Indeed, the dataset we used in the study was collected from the published literature over the past decades, and the surface water or groundwater samples were collected from different years (1964 ~ 2024) and seasons/months. To address your concerns and improve the readability of the revised manuscript, the detailed temporal dimension of the sampling year and sampling month has been supplemented in Table S1 in the ESM2.xlsx. Based on your suggestions, we have briefly explored the impact of temporal variability in water chemistry on risk assessment in the **Discussion** of the revised manuscript (**Lines 718-725**):

Temporal variations in water chemistry (e.g., seasonal fluctuations and monsoon events) significantly impact the environmental fate of contaminants and health risks through multiple mechanisms. During the monsoon season, heavy rainfall flushes tailings ponds or open-pit mines, causing instantaneous spikes in HMs (e.g., Cd, Cr, and As) and sulfate concentrations. Meanwhile, the elevated groundwater levels associated with high precipitation infiltration drive contaminant plumes along preferential pathways. These dynamics introduce systematic biases into traditional static risk assessments. The annual or quarterly average risk assessment model may underestimate short-term high-dose exposure risks. However, the temporal dimension of the dataset used in the study is not yet sufficient to further explore the above issues from a national-scale perspective. Therefore, we will provide an in-depth insight into our future studies.

---

## Author Comment (AC5)

Response to Referee #2's comments on the manuscript (HESS-2024-387)
**"Mapping mining-affected water pollution in China: Status, patterns, risks, and implications"** by Ziyue Yin, Jian Song, Dianguang Liu, Jianfeng Wu[*], Yun Yang, Yuanyuan Sun, and Jichun Wu

Note that the following text in Arial Narrow font denotes Referee's comments and in Times New Roman font denotes our response to the comments in the review.

The manuscript aims to provide a comprehensive assessment of mining-affected water pollution across China by compiling a large dataset (8433 water samples from 298 mines). The study evaluates spatial patterns, assesses both carcinogenic and non-carcinogenic risks to human health, and discusses management implications for both coal and metal mining areas. While the work is well supported by extensive data and robust methodologies, questions remain regarding the novelty of the contribution, as the manuscript does not clearly delineate how its findings significantly extend beyond previous studies.

**[Response]** We sincerely thank you for your constructive and conscientious suggestions. Hereby we have fully incorporated and addressed all the comments in the revised manuscript and given a point-by-point response as below. In particular, a more explicit statement of our novel contributions relative to the existing literature (*e.g.*, Cheng, 2003; He et al., 2013; Hu et al., 2013; Feng et al., 2014; Sun et al., 2025) has been added to the **Conclusion** to address your concern that "the manuscript does not clearly delineate how its findings significantly extend beyond previous studies".

"It is noteworthy that previous studies predominantly concentrated on localized water pollution from individual coal or metal mines, while national-scale assessments have primarily addressed impacts exclusively attributed to coal mining activities. The new and unique contributions of the current study are: (i) establishing a national-scale high-quality database covering 8,433 surface water or groundwater samples (6,175 coal mine water samples and 2,258 metal mine water samples) from 298 mines (211 coal mines and 87 metal mines) in 26 provinces/autonomous regions of China; and (ii) filling the gap of the nationwide spatial patterns of water pollution and associated health risks from both coal and metal mining activities for the first attempt. Specifically, eight heavy metals (*i.e.*, Cr, Ni, Cu, Zn, As, Cd, Hg, and Pb) are considered in the current study based on a national-scale high-quality hydrochemical database. The new results show that Zn, Ni, and Cu are the predominant contaminations contaminants of both

coal and metal mines in China. The detectable concentrations of several heavy metals are higher in most metal mines than in coal mines, especially in mining-affected water with low pH ($< 6.5$). The order of detectable median values of water affected by coal mining is Zn (0.4211) > Ni (0.1796) > Cu (0.0431) > Cr (0.0080) > Cd (0.0036) > As (0.0034) > Pb (0.0023) > Hg (0.0004), while that of water affected by metal mining is Zn (7.200) > Cu (1.7325) > Ni (0.2142) > Pb (0.1498) > Cr (0.0500) > Cd (0.0383) > As (0.0281) > Hg (0.0090). In terms of spatial patterns, the pollution hotspots and potential risks of mining-affected water (with low pH, high sulfate, Fe, Mn, and heavy metals) are pronounced in the southern regions, especially in Guizhou, Guangdong, Fujian, Jiangxi, Hunan, and Guangxi provinces/autonomous regions. These phenomena are closely linked to the underlying mechanisms, such as climatic conditions, geological factors, and mining practices. Accordingly, the findings of the study yield critical insights for designing differentiated management measures and formulating spatially-adaptive pollution control strategies across three key dimensions, including geographic scales (site-specific scale, provincial scale, or national scale), mine types (coal or metal), and mining status (active or abandoned). This multidimensional framework enables policymakers to strategically balance the trade-off between green mining activities and human health priorities."

**Cited References:**

Cheng, S.: Heavy metal pollution in China: origin, pattern and control. Environ. Sci. Pollut. Res., 10(3), 192-198. https://doi.org/10.1065/espr2002.11.141.1, 2003.

Feng, Q., Li, T., Qian, B., Zhou, L., Gao, B., and Yuan, T.: Chemical Characteristics and Utilization of Coal Mine Drainage in China. Mine Water Environ., 33, 276-286, https://doi.org/10.1007/s10230-014-0271-y, 2014.

He, B., Yun, Z.J., Shi, J.B., and Jiang, G.B.: Research progress of heavy metal pollution in China: Sources, analytical methods, status, and toxicity. Chin. Sci. Bull., 58(2), 134-140, https://doi.org/10.1007/s11434-012-5541-0, 2013.

Hu, H., Jin, Q., and Kavan, P.: A study of heavy metal pollution in China: Current status, pollution-control policies and countermeasures. Sustainability, 6, 5820-5838, https://doi.org/10.3390/su6095820, 2014.

Sun, Y.J., Guo, J., Xu, Z.M., Zhang, L., Chen, G., Xiong, X.F., Hua, J.F., Mu, L.J., and Wu, W.X.: Spatial distribution characteristics of mine water quality in coal mining areas of China and technological approaches for mine water treatment. J. China Coal Soc., 50(1), 584-599 (in Chinese with English abstract), https://doi.org/10.13225/j.cnki.jccs.YG24.1547, 2025.

**Novelty and Original Contribution**

Strengths: The study assembles a large national dataset and applies risk assessment models to evaluate health impacts, which is commendable. The spatial mapping of contamination hotspots and risk distribution provides valuable insights for policy-making.

[Response] No change needed. Thank you for your positive comments.

Concerns: One key issue is that the manuscript does not sufficiently highlight what is new compared to earlier studies. Although the scale of the data collection is impressive, the paper lacks a clear statement of its novel contributions relative to existing literature. The authors could enhance the manuscript by emphasizing unique aspects—such as new methodological approaches, previously unreported spatial trends, or innovative risk assessment strategies—that set this work apart.

[Response] The point is well taken. We have incorporated your concerns into the revision and a more explicit statement has been added to the revised **Conclusion**, highlighting the novelty and practical implications of our manuscript relative to the existing literature. See also the response to your general comment above.

**Methodological Rigor and Data Quality**

Strengths: The methodology is generally robust, with clear criteria for data quality control and appropriate use of standard risk assessment models (e.g., those provided by the US EPA). The division of water samples (acidic vs. neutral/alkaline) and the differentiation between coal and metal mines are well executed.

[Response] No change needed. Thank you for your positive comments.

Suggestions: To further strengthen the paper, the authors should elaborate on how potential biases (e.g., variations in sample density among regions) were addressed. Additionally, more detailed statistical tests comparing water quality parameters between different mining types (such as using non-parametric tests) could provide further evidence for the observed differences.

[Response] Comment accepted. Indeed, there are potential biases caused by variations in sample density among regions. Therefore, our future in-depth research will attempt to address the biases by (i) combining the data mining and field sampling methods to investigate the potential contamination levels in more coal and metal mines across China; (ii) balancing the sampling density within each zone using bias correction techniques (*e.g.*, kernel density estimation and stratified spatial resampling) to ensure the data representation; and (iii) incorporating spatial uncertainty into the criteria to improve the spatial robustness for the assessments of mining-affected

water pollution. Furthermore, we have added the results of non-parametric tests (*i.e.*, Mann-Whitney U-test and Spearman's rank correlation) to further support the differences observed in our study based on your suggestions:

"Non-parametric tests do not rely on assumptions about the distribution of the data and are suitable for non-normally distributed datasets or those containing outliers (Cardew, 2003). These methods statistically compare central tendencies, typically represented by medians, rather than means. The result of the Mann-Whitney U-test ($p < 0.05$) shows a statistically significant difference in the critical parameters (except Fe) of mining-affected water based on the different mine types (coal mine vs. metal mine), indicating the differences caused by geological factors, mining practices, surrounding environment, etc. Besides, Fig. S7 shows the Spearman correlation coefficients between the hydrochemical compositions in the mining-affected water. It can be seen that strong negative correlations are observed between pH and $SO_4^{2-}$, Fe, Mn, Al, and heavy metals while positive correlations are observed between $SO_4^{2-}$ and metal components, implying that the spatial consistency of acid water, high sulfate, high Fe and Mn, and high heavy metal mining-affected water."

[Figure]

**Figure S7.** The Spearman correlation coefficient between the hydrochemical compositions in the mining-affected water.

**Cited Reference:**

Cardew, P.T.: A method for assessing the effect of water quality changes on plumbosolvency using random daytime sampling. Water Res., 37(12), 2821-2832, https://doi.org/10.1016/S0043-1354(03)00120-9, 2003.

**Presentation and Interpretation of Results**

Strengths: The results are logically presented, starting from the basic water quality parameters, moving on to spatial distribution patterns, and culminating in detailed risk assessments for different populations. Figures (e.g., maps and boxplots) support the textual description and help visualize the trends effectively.

**[Response]** No change needed. Thank you for your positive comments.

Suggestions: Although the numerical details are extensive, the manuscript may benefit from a more concise presentation. For example, summarizing key quantitative findings in a table could improve clarity. Additionally, while the spatial patterns are well described, a deeper discussion on the underlying geochemical or environmental processes that cause these trends would better contextualize the results.

**[Response]** Thank you for your insightful comments. To improve the clarity of the manuscript, key quantitative results (*e.g.*, statistics of critical parameters for acid and neutral/alkaline water across different mines) have been summarized in Table 1 in the revised manuscript and Table S4 in the **Supplement**. Moreover, a further elaboration of the underlying mechanisms (*e.g.*, geochemical conditions and environmental processes) driving the spatial patterns of mining-affected water pollution in China, especially in the highly polluted southern regions, has been added in Section 4.1 of the revised manuscript:

[revised manuscript text omitted]

**Cited References:**

Ai Y.L., Chen, H.P., Chen, M.F., Huang, Y., Han, Z.T., Liu, G., Gao, X.B., Yang, L.H., Zhang, W.Y., Jia, Y.F., and Li, J.: Characteristics and treatment technologies for acid mine drainage from abandoned coal mines in major coal-producing countries. J. China Coal Soc., 48(12), 4521-4535 (in Chinese with English abstract), https://doi.org/10.13225/j.cnki.jccs.2022.1846, 2023.

Blowes, D.W., Ptacek, C.J., Jambor, J.L., Weisener, C.G., Paktunc, D., Gould W.D., and Johnson, D.B. The geochemistry of acid mine drainage. Treatise on Geochemistry (Second Edition), 11, 131-190, https://doi.org/10.1016/B978-0-08-095975-7.00905-0, 2014.

Jiang, C.F., Gao, X.B., Hou, B.J., Zhang, S.T., Zhang, J.Y., Li, C.C., and Wang, W.Z.: Occurrence and environmental impact of coal mine goaf water in karst areas in China. J. Cleaner Product., 275, 123813, https://doi.org/10.1016/j.jclepro.2020.123813, 2020.

Sun, Y.J., Guo, J., Xu, Z.M., Zhang, L., Chen, G., Xiong, X.F., Hua, J.F., Mu, L.J., and Wu, W.X.: Spatial distribution characteristics of mine water quality in coal mining areas of China and technological approaches for mine water treatment. J. China Coal Soc., 50(1), 584-599 (in Chinese with English abstract), https://doi.org/10.13225/j.cnki.jccs.YG24.1547, 2025.

Sun, Y.J., Zhang, L., Xu, Z.M., Chen, G., Zhao, X.M., Li, X., Gao, Y.T., Zhang, S.G., and Zhu, L.L.: Multi-field action mechanism and research progress of coal mine water quality formation and evolution. J. China Coal Soc., 47(1), 423-437 (in Chinese with English abstract), https://doi.org/10.13225/j.cnki.jccs.YG21.1937, 2022.

Tang, Y.G., He, X., Cheng, A.G., Li, W.W., Deng, X.J., Wei, Q., and Li, L.: Occurrence and sedimentary control of sulfur in coals of China. J. China Coal Soc., 40(9), 1977-1988 (in Chinese with English abstract), https://doi.org/10.13225/j.cnki.jccs.2015.0434, 2015.

**Structure and Coherence of the Argument**

Strengths: The manuscript follows a conventional structure (introduction, methodology, results, discussion, conclusion) that makes it easy to follow. The discussion ties the findings back to the broader context of water pollution management.

[Response] No change needed. Thank you for your positive comments.

Suggestions: The transition between sections—especially from the results to the discussion—could be smoother. Explicitly linking how each result addresses the stated objectives would reinforce the coherence of the argument. Also, highlighting the novelty and practical implications of the work in the conclusion would help reinforce the manuscript's contribution.

**[Response]** Comment accepted. We have rewritten the transition between sections to reinforce the coherence of the argument. Moreover, a detailed discussion of the novelty and practical implications of our manuscript has been supplemented in the **Conclusion** to highlight the unique contribution and valuable addition to the field of environmental hydrology. See also the response to your general comment above.

**Figures, Tables, and Visual Aids**

Strengths: Visual aids are generally clear and provide a good overview of the data distribution and risk maps. The integration of detailed figures (such as spatial distribution maps and risk assessment graphs) adds significant value to the manuscript.

**[Response]** No change needed. Thank you for your positive comments.

Suggestions: Ensure that all figures have clear legends and consistent formatting. It might be beneficial to include a summary table that aggregates the key findings (e.g., median values of critical parameters across different mine types) to enhance readability.

**[Response]** Thank you for your constructive suggestions. We have reviewed/revised all figures to ensure that they have clear legends and consistent formatting. To improve overall readability, the summary tables (*i.e.*, Table 1 in the revised manuscript and Table S4 in the **Supplement**) showing the statistics of critical parameters for different mine types have been added to the current revision. See also the response to your comment on 'Presentation and Interpretation of Results' above.

**Language and Style**

Strengths: The manuscript is written in clear, professional English with an appropriate academic tone. Technical terms are defined upon first use, and the text is generally free of major grammatical errors.

**[Response]** No change needed. Thank you for your positive comments.

Suggestions: A few sentences could be simplified to improve readability. In particular, some complex sentences in the introduction and discussion might be broken into shorter, more digestible statements. Maintaining consistency in terminology (for instance, ensuring that terms like "differentiated management" are clearly defined) will also help in reinforcing the manuscript's clarity.

**[Response]** Comment accepted. We have simplified some complex sentences in the **Introduction** and **Discussion** to improve the readability of the manuscript. Moreover, a clear definition of terms

like "differentiated management" has been added in the revised manuscript to reinforce the manuscript's clarity:

"The differentiated management mentioned in the current study is an optimized regulatory paradigm that customizes strategies to mine types (coal vs. metal) and operational status (active vs. abandoned) based on hydrogeological conditions, pollution source characteristics, and multi-system sustainability requirements. The initiative aims to implement targeted intervention and precise prevention/control to mitigate pollution risks, restore and enhance ecological functions, while concurrently safeguarding human health."

**Conclusion**

The manuscript presents an extensive dataset and a rigorous analysis of mining-affected water pollution in China, offering useful insights for environmental management and policy-making. However, the work would benefit from a more explicit discussion of its novelty compared to previous studies. Clarifying and emphasizing the unique contributions—whether in data scale, methodological advancements, or new insights into spatial and health risk patterns—would significantly strengthen the paper. With these revisions, the manuscript could represent a valuable addition to the field of environmental hydrology.

**[Response]** We sincerely appreciate your conscientious and constructive comments. A more explicit discussion of the novelty compared to previous studies has been added to the revised manuscript, to provide new insights into the spatial patterns and health risks of mining-affected water pollution at the national scale, and to clarify and emphasize the unique contributions of our study. We believe that your insightful comments on 'Novelty and Original Contribution', 'Methodological Rigor and Data Quality', 'Presentation and Interpretation of Results', 'Structure and Coherence of the Argument', 'Figures, Tables, and Visual Aids', and 'Language and Style' have led to significant improvements of the revised manuscript.

---

## Author Comment (AC6)

Response to Referee #3's comments on the manuscript (HESS-2024-387)
**"Mapping mining-affected water pollution in China: Status, patterns, risks, and implications"** by Ziyue Yin, Jian Song, Dianguang Liu, Jianfeng Wu[*], Yun Yang, Yuanyuan Sun, and Jichun Wu

Note that the following text in Arial Narrow font denotes Referee's comments and in Times New Roman font denotes our response to the comments in the review.

The study delivers a thorough and spatially explicit examination of mining-induced water pollution and associated health risks across China, utilizing an extensive dataset comprising 8,433 samples. The differentiation between coal and metal mines, along with the identification of southern China as a pollution hotspot, provides valuable insights for region-specific policy formulation. Below are several constructive suggestions for refining the manuscript:

**[Response]** We sincerely appreciate your constructive comments and the recommendation for minor revisions of the manuscript. Moreover, we have made the necessary changes to the original manuscript and hereinafter provided a point-by-point response.

1. While the spatial heterogeneity of pollution is convincingly presented, the underlying mechanisms driving the pronounced contamination in southern China (e.g., geological factors, mining practices, or climatic conditions) warrant further elaboration. Incorporating a brief discussion that connects regional geochemistry or historical mining activities to observed pollution patterns would enhance the robustness of the analysis.

**[Response]** Thank you for your positive and constructive suggestions. As you suggested, a further elaboration of the underlying mechanisms (*e.g.*, geological factors, mining practices, and climatic conditions) driving the pronounced contamination in South China has been added in Section 4.1 of the revised manuscript. Furthermore, to enhance the robustness of the analysis, the potential relation between the observed water pollution patterns and both regional geochemical characteristics and historical mining activities has been discussed in detail as follows:

"The underlying mechanisms, including climatic conditions, geological factors, and mining practices, determine the spatial patterns of mining-affected water pollution in China, especially in the highly polluted southern regions. In terms of climatic conditions, the average temperature of the coldest month is > 0°C, while that of the hottest month is > 22°C, and the annual average

precipitation is generally > 1,000 mm in South China. The high temperature and precipitation create a synergistic accelerator for mine water acidification. Elevated temperatures stimulate acidophilic microbial communities (*e.g.*, *Acidithiobacillus ferrooxidans*), which enhance enzymatic activity that catalyzes sulfide mineral oxidation. Combined with high levels of precipitation, rainfall infiltrates abandoned mines, tailings ponds, and exposed ore bodies, creating a sustained water-oxygen exchange that drives sulfuric acid formation and iron oxidation processes."

"In terms of geological factors, the unique geo-environmental settings of South China, characterized by rugged topography, widespread sulfur-rich strata, and high background value of metallic minerals, result in mining-affected water with high acidity and elevated concentrations of sulfate, Fe, Mn, and HMs (Sun et al., 2022). The coal-forming periods of different mines in the South China coalfields are diverse, mainly Triassic, Neoproterozoic, etc., of which the sulfur enrichment exhibits strong links to marine-land interactions. The sustained seawater intrusion-regression cycle results in elevated sulfur contents (predominantly medium and high-sulfur coals) (Ai et al., 2023; Sun et al., 2025). As illustrated in Fig. S2, the coal fields in China exhibit sulfur contents ranging from 0.02% to 10.48%, with South China's coal-bearing areas showing the highest weighted average sulfur content (2.35%), including 29.63% of high-sulfur coal. Comparatively, those weighted average sulfur contents of coal-bearing areas in Tibet-Western Yunnan, North China, and Northeast China are 0.94%, 0.88%, and 0.86%, respectively (Tang et al., 2015). In addition, as shown in Fig. S3, the metal mineral resources are abundant in the southern region of China, and the water affected by mining practices is often highly toxic, with harmful HMs such as Cd, Pb, Hg, Cr, As, Cu, and so on, endangering the surface water and groundwater systems (Sun et al., 2022)."

"As to mining practices, especially those involving sulfide-bearing metalliferous ore deposits and sulfide-rich coal mining, are intrinsically associated with AMD. Acid drainage can occur wherever sulfide minerals are excavated and exposed to atmospheric oxygen. The main sulfide minerals in mine wastes are pyrite ($FeS_2$) and pyrrhotite ($Fe_{1-x}S$), while other associated sulfides are prone to oxidation and release toxic elements, including Al, As, Cd, Co, Cu, Hg, Ni, Pb and Zn, into the water flowing through the mine tailings (Blowes et al., 2014). Moreover, underground mining is the primary exploitation method in China. Substantial mined-out areas are formed after mining activities, inducing the accumulation of groundwater and the formation of acid mine water. In recent years, the phenomenon has intensified because a number of mines are abandoned without proper closure measures (Jiang et al., 2020)."

[Figure]

**Figure S2.** The total sulfur content in different coal-bearing areas in China (adapted from Tang et al., 2015).

[Figure]

**Figure S3.** The spatial distribution of the major non-ferrous mineral resources in China (adapted from China Natural Resources Atlas, China Geological Survey, 2015, https://www.cgs.gov.cn/xwl/dzzl/201603/t20160309_304269.html).

**Cited References:**

Ai Y.L., Chen, H.P., Chen, M.F., Huang, Y., Han, Z.T., Liu, G., Gao, X.B., Yang, L.H., Zhang, W.Y., Jia, Y.F., and Li, J.: Characteristics and treatment technologies for acid mine drainage from abandoned coal mines in major coal-producing countries. J. China Coal Soc., 48(12), 4521-4535 (in Chinese with English abstract), https://doi.org/10.13225/j.cnki.jccs.2022.1846, 2023.

Blowes, D.W., Ptacek, C.J., Jambor, J.L., Weisener, C.G., Paktunc, D., Gould W.D., and Johnson, D.B. The geochemistry of acid mine drainage. Treatise on Geochemistry (Second Edition), 11, 131-190, https://doi.org/10.1016/B978-0-08-095975-7.00905-0, 2014.

Jiang, C.F., Gao, X.B., Hou, B.J., Zhang, S.T., Zhang, J.Y., Li, C.C., and Wang, W.Z.: Occurrence and environmental impact of coal mine goaf water in karst areas in China. J. Cleaner Product., 275, 123813, https://doi.org/10.1016/j.jclepro.2020.123813, 2020.

Sun, Y.J., Guo, J., Xu, Z.M., Zhang, L., Chen, G., Xiong, X.F., Hua, J.F., Mu, L.J., and Wu, W.X.: Spatial distribution characteristics of mine water quality in coal mining areas of China and technological approaches for mine water treatment. J. China Coal Soc., 50(1), 584-599 (in Chinese with English abstract), https://doi.org/10.13225/j.cnki.jccs.YG24.1547, 2025.

Sun, Y.J., Zhang, L., Xu, Z.M., Chen, G., Zhao, X.M., Li, X., Gao, Y.T., Zhang, S.G., and Zhu, L.L.: Multi-field action mechanism and research progress of coal mine water quality formation and evolution. J. China Coal Soc., 47(1), 423-437 (in Chinese with English abstract), https://doi.org/10.13225/j.cnki.jccs.YG21.1937, 2022.

Tang, Y.G., He, X., Cheng, A.G., Li, W.W., Deng, X.J., Wei, Q., and Li, L.: Occurrence and sedimentary control of sulfur in coals of China. J. China Coal Soc., 40(9), 1977-1988 (in Chinese with English abstract), https://doi.org/10.13225/j.cnki.jccs.2015.0434, 2015.

2. The health risk assessment (e.g., 51.52% carcinogenic risk for adults) raises significant concerns but lacks sufficient methodological detail. Please specify the exposure parameters employed (e.g., ingestion rates, body weight assumptions) and the toxicity thresholds applied. Additionally, clarify whether the identified risks stem from specific contaminants (e.g., arsenic, cadmium) or synergistic interactions among multiple pollutants.

**[Response]** Comment accepted. The main parameters used for the human health risk assessment (*e.g.*, ingestion rates, exposure frequency, body weight, etc.) are presented in Table S2. The values of permeability coefficient of skin ($K_P$), reference dose ($RfD_o$), gastrointestinal digestion coefficient ($ABS_{GI}$), and slope factor (SF) for each element are described in Table S3 in Section S3 of the **Supplement**.

**Table S2.** The main parameters used for human health risk assessment.

| Parameter | Description | Unit | Value Adult | Value Children | Source |
|---|---|---|---|---|---|
| *IR* | Ingestion rate | L/d | 2.50 | 0.78 | [1], [2] |
| *EF* | Exposure frequency | d/yr | 350 | 350 | [1], [2] |
| *ED* | Exposure duration | yr | 24 | 6 | [2] |
| *ET* | Time of contact | h/d | 0.58 | 1.00 | [3], [4] |
| *SA* | Skin surface area | cm$^2$ | 19652 | 6365 | [1], [2] |
| *CF* | Conversion factor | L/cm$^3$ | 0.001 | 0.001 | [2], [5] |
| *BW* | Body weight | kg | 70 | 15 | [1], [3], [4] |
| *AT* | Averaging time [a] | d | 8760 | 2190 | $ED \times 365$ d/yr |
| | Averaging time [b] | d | 25550 | 25550 | $70 \times 365$ d/yr |

Note: [a] averaging time used for non-carcinogenic risks (NCRs), and [b] averaging time used for carcinogenic risks (CRs), which is equal to a lifetime (70 yr in the study) ×365 d/yr. The parameter values used in the study are obtained from the following literature sources: [1] Meng et al. (2024); [2] Shi et al. (2023); [3] Tong et al. (2021); [4] Wang et al. (2021); and [5] Yuan et al. (2023).

**Table S3.** The values of main parameters including permeability coefficient of skin ($K_p$), reference dose ($RfD_o$), gastrointestinal digestion coefficient ($ABS_{GI}$), and slope factor (SF) for each element.

| Parameter | $K_p$ (cm/h) | $RfD_o$ (mg/kg·d) | $ABS_{GI}$ (-) | SF (kg·d/mg) | Source |
|---|---|---|---|---|---|
| Fe | 0.001 | 0.7 | 0.2 | - | [1], [2], [3], [4], [6] |
| Mn | 0.001 | 0.024 | 0.04 | - | [1], [2], [3], [4], [6] |
| Cr | 0.002 | 0.003 | 0.025 | 0.5 | [1], [3], [6], [7] |
| Ni | 0.0002 | 0.02 | 0.04 | - | [1], [2], [3], [4], [6], [7] |
| Cu | 0.001 | 0.04 | 0.2 | - | [1], [2], [3], [4], [6], [7] |
| Zn | 0.0006 | 0.3 | 0.2 | - | [1], [2], [3], [4], [5], [6] |
| As | 0.001 | 0.0003 | 1 | 1.5 | [1], [3], [7] |
| Cd | 0.001 | 0.0005 | 0.05 | 0.38 | [2], [3], [4], [6] |
| Pb | 0.0001 | 0.0014 | 0.3 | - | [1], [3], [6] |

Note: The parameter values for each element are obtained from the following literature sources: [1] Meng et al. (2024); [2] Shi et al. (2023); [3] Tong et al. (2021); [4] USEPA (2002); [5] USEPA (2014); [6] Wang et al. (2021); and [7] Zheng et al. (2023).

In the study, Cr, Cd, and As are considered for the calculation of the carcinogenic risks. On the other hand, Fe, Mn, Cr, Ni, Cu, Zn, As, Cd, and Pb are taken into consideration to identify the non-carcinogenic risks. Note that the identified risks (*e.g.*, 51.52% carcinogenic risk for adults) stem from synergistic interactions among multiple pollutants, which have been clarified in the revised manuscript to avoid unnecessary misunderstandings:

"In connection with the results displayed in Fig. S9a, the mining areas with non-negligible CRs (TCR > $10^{-4}$), primarily driven by the combined effects of Cr, Cd, and As, account for 68.25 % of adults and 51.47% of children exposed to T1-type water, and 40.27% of adults and 23.31% of children exposed to T2-type water."

" In connection with the results displayed in Fig. S9b, the mining areas with high HI values (HI > 1, stemming from synergistic interactions among Fe, Mn, Cr, Ni, Cu, Zn, As, Cd, and Pb) account for 88.35 % (for adults) and 91.90% (for children) exposed to T1-type water. For T2-type water, the corresponding proportions are 55.75% (for adults) and 63.10% (for children)."

[Figure]

**Figure S9.** The cumulative distribution function (CDF) of (a) total carcinogenic risk (TCR) and (b) hazard index (HI) in mining-affected water. T1 category includes mine drainage, mine water, and leachate water, while T2 category indicates mining-affected surface water and groundwater.

**Cited References:**

Meng, F., Cao, R., Zhu, X., Zhang, Y., Liu, M., Wang, J., Chen, J., and Geng, N.: A nationwide investigation on the characteristics and health risk of trace elements in surface water across China. Water Res., 250, 121076, https://doi.org/10.1016/j.watres.2023.121076, 2024.

Shi, J., Zhao, D., Ren, F., and Huang, L.: Spatiotemporal variation of soil heavy metals in China: The pollution status and risk assessment. Sci. Total Environ., 871, 161768, https://doi.org/10.1016/j.scitotenv.2023.161768, 2023.

Tong, S., Li, H., Tudi, M., Yuan, X., and Yang, L.: Comparison of characteristics, water quality and health risk assessment of trace elements in surface water and groundwater in China. Ecotox. Environ. Safe., 219, 112283, https://doi.org/10.1016/j.ecoenv.2021.112283, 2021.

USEPA: Risk-based Concentration Table. U.S. Environment Protection Agency (Washington DC), 2002.

USEPA: Human health evaluation manual, supplemental guidance: update of standard default exposure factors. Environment Protection Agency (Washington DC), 2014.

Wang, J., Liu, G., Liu, H., and Lam, P.K.S.: Multivariate statistical evaluation of dissolved trace elements and a water quality assessment in the middle reaches of Huaihe River, Anhui, China. Sci., Total Environ., 583, 421-431, https://doi.org/10.1016/j.scitotenv.2017.01.088, 2017.

Yuan, R., Li, Z., and Guo, S.: Health risks of shallow groundwater in the five basins of Shanxi, China: Geographical, geological and human activity aspects. Environ. Pollut., 316, 120524, https://doi.org/10.1016/j.envpol.2022.120524, 2023.

Zheng, X., Lu, Y., Xu, J., Geng, H., and Li, Y.: Assessment of heavy metals leachability characteristics and associated risk in typical acid mine drainage (AMD)-contaminated river sediments from North China. J. Clean. Product., 413, 137338, https://doi.org/10.1016/j.jclepro.2023.137338, 2023.

3. The temporal dimension of water sampling remains ambiguous. Were the samples collected across different seasons or years? Temporal variability in water chemistry (e.g., the influence of monsoon events on metal mobility) could significantly affect risk estimates and merits further exploration in the discussion section.

**[Response]** We appreciate your insight. Indeed, the dataset we used in the study was collected from the published literature over the past decades, and the surface water or groundwater samples were collected from different years (1964 ~ 2024) and seasons/months. To address your concerns and improve the readability of the revised manuscript, the detailed temporal dimension of the sampling year and sampling month has been supplemented in Table S1 in the ESM2.xlsx. Based on your suggestions, we have briefly explored the impact of temporal variability in water chemistry on risk assessment in the **Discussion** of the revised manuscript:

"Temporal variations in water chemistry (e.g., seasonal fluctuations and monsoon events) significantly impact the environmental fate of contaminants and health risks through multiple mechanisms. During the monsoon season, heavy rainfall flushes tailings ponds or open-pit mines, causing instantaneous spikes in HMs (e.g., Cd, Cr, and As) and sulfate concentrations. Meanwhile, the elevated groundwater levels associated with high precipitation infiltration drive contaminant plumes along preferential pathways. These dynamics introduce systematic biases into traditional static risk assessments. The annual or quarterly average risk assessment model may underestimate

short-term high-dose exposure risks. However, the temporal dimension of the dataset used in the study is not yet sufficient to further explore the above issues from a national-scale perspective. Therefore, we will provide an in-depth insight into our future studies."

---

## Author Response (AR2)

Response to the comments on the manuscript (HESS-2024-387) **"Mapping mining-affected water pollution in China: Status, patterns, risks, and implications"** by Ziyue Yin, Jian Song, Dianguang Liu, Jianfeng Wu[*], Yun Yang, Yuanyuan Sun, and Jichun Wu.

Note that the following text in Arial Narrow font denotes Editor's comments and in Times New Roman font denotes our response to the comments in the review. In our resubmission, the marked PDF file (hess-2024-387_ATC2.pdf) has clearly indicated all changes to the original manuscript, tables and figures. Also, in our marked PDF file, marked in a green strikethrough font is the text that should be removed from the original manuscript and marked in a red font is the text that has been added to the current revision. In addition, Line number(s) mentioned below can be referred to as that line numbering in the marked revised manuscript.

**Response to Gabriel Rau's Comments**

Dear authors, thank you for thoroughly revising this manuscript. The reviewers agree that you have done an excellent job. However, I do have a few minor issues that I would like to see addressed before I can accept the manuscript for publication in HESS.

**[Response]** Dear Prof. Gabriel Rau, we sincerely thank you for your positive feedback and the opportunity to further improve our manuscript. We have fully incorporated and addressed all the comments in the revised manuscript and given a point-by-point response as below. In addition, following the editorial support team's recommendations, we have revised the reference list to align with the journal's manuscript preparation guidelines. We believe that the constructive and conscientious comments have led to significant improvement of the revised manuscript.

All figure and table captions (except for Figure 1?) need to be expanded so that they describe to the reader what is being shown. This is to help speed readers digest the materials. For example, Figure 2 should read: "Statistical summary (minimum, median, average and maximum) of the main species aggregated from all samples measured in mining-affected water in China (Units are mg/L except pH)."

**[Response]** Thank you for your constructive suggestions. As you suggested, we have expanded captions to all the figures and tables in both the main manuscript and the supplement to enhance clarity and facilitate reader comprehension. For your convenience, the expanded captions in the revised manuscript are successively listed below:

**Figure Captions**

**Figure 2.** Statistical summary (minimum, median, average, and maximum) of the main species aggregated from all samples measured in mining-affected water in China (Units are mg/L except pH).

**Figure 3.** The relationship between pH and the respective concentrations including (a) $SO_4^{2-}$, (b) Fe, (c) Mn, and (d) Al, in coal and metal mines. The binned frequency distribution of the samples is shown along the x and y axes.

**Figure 4.** The spatial distribution of (a) pH; and the mean concentration of individual components (mg/L) showing (b) $SO_4^{2-}$, (c) Fe, and (d) Mn, respectively, in mining-affected water on the 0.5° grid. The classification thresholds for the main components are based on the distribution of all collected data, as well as regulatory benchmarks from GB 3838-2002 and GB/T 14848-2017 in China.

**Figure 5.** The carcinogenic risk (CR) values of Cr, As, and Cd in mining-affected water. T1 category includes mine drainage, mine water, and leachate water, while T2 category indicates mining-affected surface water and groundwater.

**Figure 6.** The hazard quotient (HQ) values of Fe, Mn, Cr, Ni, Cu, Zn, As, Cd, and Pb in mining-affected water for (a) T1-Adult, (b) T1-Children, (c) T2-Adult, and (d) T2-Children, respectively. T1 category includes mine drainage, mine water, and leachate water, while T2 category indicates mining-affected surface water and groundwater.

**Figure 7.** Conceptual model illustrating the pollution pathways and environmental impacts of mining-affected water across four key subsystems: (i) groundwater, (ii) surface water, (iii) soil, and (iv) human health, during both active mining and abandoned phases.

**Table Caption**

**Table 1.** Statistical summary (minimum, median, average, and maximum) of the main species aggregated from all samples measured in acid or non-acid mining-affected water in China (Units are mg/L except pH).

The abstract needs to be tweaked to include a few words that you have calculated quantitative indicators as described in Section 2.4.

**[Response]** Comment accepted. We have revised the abstract to explicitly include the quantitative risk assessment results from Section 2.4 (**Line 24**), as follows:

"The risk assessment reveals that the unacceptable carcinogenic risks caused by poor-quality surface water and groundwater are observed in 51.52% (for adults) and 29.29% (for children) of the mining areas.

Moreover, severe non-carcinogenic risks are also identified in 68.07% and 80.67% of mining areas for adults and children, respectively."

**[Response]** We sincerely thank you for your insightful suggestions. We have moved the $SO_4^{2-}$ plot to the right side of Figure 2. Additionally, clear y-axis labels have been added to all subplots to improve readability.

[Figure]

**Figure 2.** Statistical summary (minimum, median, average, and maximum) of the main species aggregated from all samples measured in mining-affected water in China (Units are mg/L except pH).

Table 1: Please make sure that the reported digits for concentrations (mg/L) represent what can realistically be measured. For example, there are a lot of trailing zeros that could be rounded.

**[Response]** We appreciate your comment regarding the number of reported digits for concentration values (mg/L). We confirm that the reported concentrations can realistically be measured. In our study, the composite database integrates mining-affected water (surface water and groundwater) quality parameters systematically extracted from 293 peer-reviewed studies published over the past decades. Due to inconsistencies in the original reporting units (some sources used mg/L, while others used μg/L), we standardized all concentration values to mg/L to ensure comparability across the dataset. As a result of this unit conversion, particularly for trace elements originally reported in μg/L, some values contain up to four decimal places. To address your concern, we have rounded the mean and maximum values of heavy metal concentrations to two significant figures. However, the minimum and median values are still presented with up to four decimal places to preserve the resolution of trace-level concentrations, particularly for values close to the detection limit.

Figure 3: Add a note to the caption to state that the binned frequency distribution (?) of samples is shown along the x and y axes.

[Response] Comment accepted. The caption of Figure 3 has been revised to "The respective relationship between pH and the concentrations of (a) $SO_4^{2-}$, (b) Fe, (c) Mn, and (d) Al in coal and metal mines. The binned frequency distribution of the samples is shown along the x and y axes"

Figure 4: Could you please at some longitude and latitude tick marks and labels to this map.

[Response] Comment accepted. Following your valuable suggestions, we have added longitude and latitude tick marks with corresponding labels to Figure 4. For consistency, we have also incorporated these geographic coordinates in Figure 1 and supplementary Figures S1, S3, S4, S8, S10, and S11.

[Figure]

**Figure 4.** The spatial distribution of (a) pH; and the mean concentration of individual components (mg/L) showing respective (b) $SO_4^{2-}$, (c) Fe, and (d) Mn in mining-affected water on the 0.5° grid. The classification thresholds for the main components are based on the distribution of all collected data, as well as regulatory benchmarks from GB 3838-2002 and GB/T 14848-2017 in China.

Figure 5: Spell out the y axis label as "Carcinogenic Risk (CR)". This helps speed readers immediately understand without having to dig into the text.

**[Response]** Comment accepted. To improve the readability of the manuscript, we have modified the y-axis label from the abbreviated form "CR" to the full designation "Carcinogenic Risk (CR)" in Figure 5.

[Figure]

**Figure 5.** The carcinogenic risk (CR) values of Cr, As, and Cd in mining-affected water. T1 category includes mine drainage, mine water, and leachate water, while T2 category indicates mining-affected surface water and groundwater.

Same with Figure 6, spell out HQ for each of the axes.

**[Response]** Similarly, we have modified the x-axis label from the abbreviated form "HQ" to the full designation "Hazard Quotient (HQ)" in Figure 6.

Could you please spell out abbreviations upon first use in the Conclusions. Again, this helps speed readers obtain the gist of the study without having to delve into the text.

**[Response]** Thank you for your constructive suggestions. We have revised the abbreviated form "HMs" to the full designation "heavy metals" in the **Conclusions** to help readers obtain the gist of the study (**Lines 630 and 637**).

[Figure]

**Figure 6.** The hazard quotient (HQ) values of Fe, Mn, Cr, Ni, Cu, Zn, As, Cd, and Pb in mining-affected water for (a) T1-Adult, (b) T1-Children, (c) T2-Adult, and (d) T2-Children, respectively. T1 category includes mine drainage, mine water, and leachate water, while T2 category indicates mining-affected surface water and groundwater.